# Malaria parasites regulate intra-erythrocytic development duration via serpentine receptor 10 to coordinate with host rhythms

Amit K. Subudhi[1], Aidan J. O'Donnell[2,9], Abhinay Ramaprasad[1,9], Hussein M. Abkallo [3], Abhinav Kaushik [1], Hifzur R. Ansari [1], Alyaa M. Abdel-Haleem[1,8], Fathia Ben Rached [1], Osamu Kaneko [4], Richard Culleton [3,5✉], Sarah E. Reece [2✉] & Arnab Pain [1,6,7✉]

Malaria parasites complete their intra-erythrocytic developmental cycle (IDC) in multiples of 24 h suggesting a circadian basis, but the mechanism controlling this periodicity is unknown. Combining in vivo and in vitro approaches utilizing rodent and human malaria parasites, we reveal that: (i) 57% of *Plasmodium chabaudi* genes exhibit daily rhythms in transcription; (ii) 58% of these genes lose transcriptional rhythmicity when the IDC is out-of-synchrony with host rhythms; (iii) 6% of *Plasmodium falciparum* genes show 24 h rhythms in expression under free-running conditions; (iv) Serpentine receptor 10 (SR10) has a 24 h transcriptional rhythm and disrupting it in rodent malaria parasites shortens the IDC by 2-3 h; (v) Multiple processes including DNA replication, and the ubiquitin and proteasome pathways, are affected by loss of coordination with host rhythms and by disruption of SR10. Our results reveal malaria parasites are at least partly responsible for scheduling the IDC and coordinating their development with host daily rhythms.

[1] Pathogen Genomics Group, BESE Division, King Abdullah University of Science and Technology (KAUST), Thuwal 23955-6900, Kingdom of Saudi Arabia. [2] Institute of Evolutionary Biology, and Institute of Immunology and Infection Research, University of Edinburgh, Edinburgh EH9 3FL, UK. [3] Malaria Unit, Department of Pathology, Institute of Tropical Medicine (NEKKEN), Nagasaki University, 1-12-4 Sakamoto, Nagasaki 852-8523, Japan. [4] Department of Protozoology, Institute of Tropical Medicine (NEKKEN), Nagasaki University, 1-12-4 Sakamoto, Nagasaki 852-8523, Japan. [5] Division of Molecular Parasitology, Proteo-Science Center, Ehime University, 454 Shitsukawa, Toon, Ehime 791-0295, Japan. [6] Center for Zoonosis Control, Global Institution for Collaborative Research and Education (GI-CoRE), Hokkaido University, N20 W10 Kita-ku, Sapporo 001-0020, Japan. [7] Nuffield Division of Clinical Laboratory Sciences (NDCLS), University of Oxford, Headington, Oxford OX3 9DU, UK. [8] Present address: Computational Bioscience Research Center, King Abdullah University of Science and Technology (KAUST), Thuwal 23955-6900, Kingdom of Saudi Arabia. [9] These authors contributed equally: Aidan J. O'Donnell, Abhinay Ramaprasad. ✉email: richard@nagasaki-u.ac.jp; sarah.reece@ed.ac.uk; arnab.pain@kaust.edu.sa

Many parasite species exhibit daily rhythms in behavior and/or development that appear scheduled to optimally exploit periodicities in transmission opportunities and/or resource availability[1,2]. The parasitic protozoan *Trypanosoma brucei*, for example, possesses an intrinsic circadian clock that drives metabolic rhythms[3]. Similarly, rhythms in host feeding and innate immune responses influence the timing of rhythms in the intra-erythrocytic developmental cycle (IDC) of rodent malaria parasites[4,5]. Specifically, completion of the IDC, a glucose-demanding process, coincides with host food intake, and quiescence during the early phase of the IDC coincides with the daily nadir in host blood glucose that is exacerbated by the energetic demands of immune responses mounted during malaria infection[4]. However, the extent to which malaria parasites or their hosts are responsible for the synchrony, timing, and duration of the IDC schedule is unclear[6]. Either parasites are able to respond to time-of-day cues provided by the host to organize when they transition between IDC stages and complete schizogony, or, parasites are intrinsically arrhythmic and allow the host to impose rhythms on the IDC that benefit the parasite[1].

Establishing how the timing and synchronicity of the IDC is established is important because temporal coordination with host rhythms is beneficial for parasite fitness[7,8], and because tolerance to antimalarial drugs is conferred to parasites that pause their IDCs[9–11]. Here, we use a combination of rodent malaria parasites in vivo and human malaria parasites in vitro to investigate the relationship between the IDC and host circadian rhythms. First, we identify components of the *P. chabaudi* transcriptome with 24 h periodicities and determine what happens to them, including the downstream biological processes, when coordination with host rhythms is disrupted (i.e., when the parasites' IDC is out of phase with the host). Second, we show that *P. falciparum* also has a transcriptome with 24 h periodicity, even in the absence of host rhythms. Third, we identify a transmembrane serpentine receptor with 24 h rhythmic expression in both species and demonstrate it plays a role in the duration of the IDC. Furthermore, loss of this serpentine receptor disrupts many of the same processes affected when coordination to host rhythms is perturbed. Taken together, our results imply that malaria parasites are, at least in part, able to control the schedule of the IDC.

## Results

### The *P. chabaudi* transcriptome responds to host rhythms.
Transcriptome analyses of time-series RNA sequencing datasets were performed with parasites from infections that were in synchrony (phase aligned; "host rhythm matched") and 12 h out of synchrony (out-of-phase; "host rhythm mismatched") with host daily rhythms (for details see Methods section) (Fig. 1a). After mismatch to host daily rhythms, the IDC of *P. chabaudi* becomes rescheduled to match the host's rhythms. By the time of sampling (days 4–5 post-infection, PI), schizogony of mismatched parasites peaked 6 h after matched parasites (inferred from ring stage rhythms) (Supplementary Fig. 1a). Parasites in both matched and mismatched infections remained synchronous throughout the sampling period (Supplementary Fig. 1a). After quantifying gene expression at each time point through RNAseq analysis ($n = 2$ per time point), we identified genes that followed ~24 h ("daily") rhythms in expression according to two commonly used and independent algorithms (see Methods section) with a threshold of $p < 0.05$. Of a total of 5343 genes (5158 detected and considered for analysis), 3057 (58%) in matched parasites, and 1824 (34%) in mismatched parasites, exhibited daily rhythms in expression ($p < 0.05$; Supplementary Fig. 1b, c and Supplementary Data 1). A permutation test was performed to empirically determine the false discovery rate (FDR) in detecting daily rhythmic transcripts

(see Methods section). When the sampling order was permuted 1000 times, it always gave a smaller number of rhythmic genes compared to the number of observed rhythmic genes predicted when the samples were kept in correct order, with an overall FDR of <0.05 for all permutation tests conducted for each algorithm used for each experimental condition (Supplementary Fig. 1d). A similar approach was used by Rijo-Ferreira et al.[3]. We also calculated the Z-score to identify how far the observed value was from the values obtained in the permutation tests. Z-scores were found to be high, at between 5.50–16.68, for all four permutation tests done. This suggests that our observations are higher than the distribution of all permutations. When we took a range of different *q*-value cut-offs, we observed that there were always a greater number of rhythmic transcripts detected in the matched parasites compared to the mismatched parasites (Supplementary Fig. 1e).

### Effects of mismatch to the host rhythm.
Over 80% of the genes expressed during the IDC of malaria parasites undergo a tight temporal expression cascade associated with each parasite stage[12,13]. The periodicity of the expression profile of these genes can be ~24, 48, or 72 h depending on the malaria parasite species: for *P. chabaudi*, the expression profile of these genes have a ~24 h periodicity while for *P. falciparum* they have ~48 h periodicity. Thus, genes associated with IDC progression should peak 6 h later in mismatched compared to matched parasites, but any genes directly sensitive to the time-of-day of hosts should peak at the same point in the host's daily rhythm according to its light: dark schedule (i.e., ZT), which corresponds to 12 h GMT apart in mismatched compared to matched parasites (because their hosts were kept in opposite light:dark schedules).

Comparing rhythmic transcripts detected in both matched and mismatched parasites revealed three sets of transcripts: (1) 1765 genes (33% of the total) with 24 h rhythmic expression ($p < 0.05$) in matched parasites that exhibit a reduction in rhythmicity of expression profile in mismatched parasites ($p > 0.05$); (2) 1292 genes with expression profiles with 24 h rhythms in both matched and mismatched parasites; and (3) 532 genes whose expression profiles were significantly more rhythmic in mismatched than matched parasites ($p < 0.05$, Fig. 1b, c). Hierarchical clustering analysis identified biological replicates to be tightly clustered (Supplementary Fig. 2a). Comparison of the 11 time points using principal component analysis identified the first component in both the conditions with a cyclic pattern that accounted for >85% of total variance (Supplementary Fig. 2b), which supports the hypothesis that the large number of periodic transcripts are expressed with 24 h (daily) periodicity.

Out of 1292 genes that retained 24 h rhythmicity in both matched and mismatched parasites, we found 685 genes (53%) that had a delay of about 6 h (±1.5 h) in mismatched compared to matched parasites (Supplementary Fig. 2c), complementing the phase difference between their IDCs detected by morphology (Supplementary Fig. 1a). Gene ontology enrichment analysis revealed that biological processes associated with these transcripts include DNA metabolic processes and cellular responses to stress (FDR < 0.05). The other 607 genes display broad differences in their phase of expression between matched and mismatched parasites. Genes that alter their rhythmic expression in mismatched parasites may do so because the IDC and/or homeostasis are negatively affected by misalignment with host rhythms, and/or as a consequence of actively rescheduling the IDC to become coordinated to host daily rhythms.

The amplitude of rhythmically expressed genes is significantly higher ($p < 0.0001$, unpaired Student's *t*-test) in matched compared to mismatched parasites (Fig. 1d, Supplementary

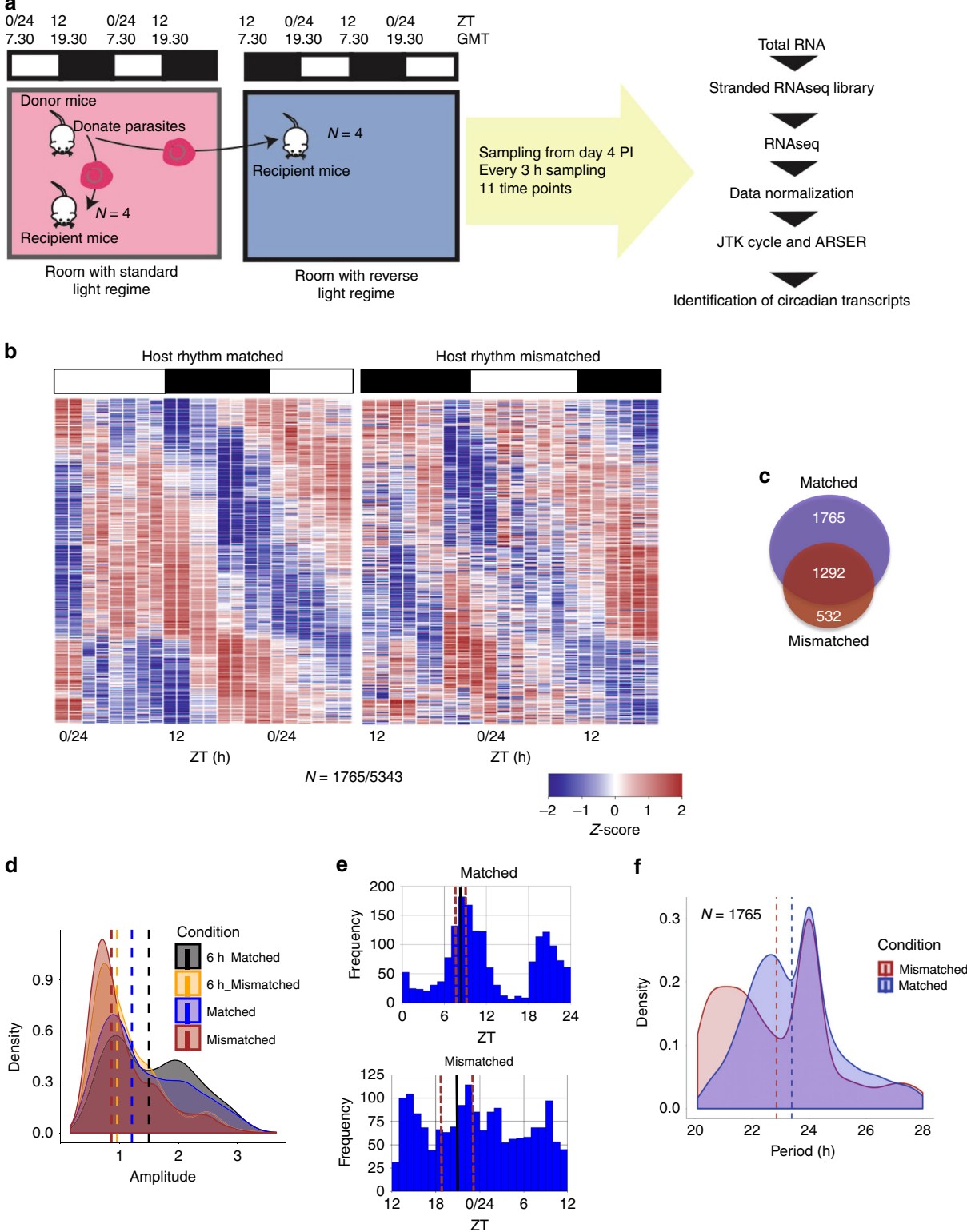

Fig. 2d), suggesting a loss of synchronicity in the gene expression of mismatched parasites that is not severe enough to impact on IDC synchronicity as measured by stage proportions (Supplementary Fig. 1a). Such a dampening of rhythms is a typical consequence of misalignment of a circadian clock with its time-of-day cue ("Zeitgeber")[14]. Genes ($N = 532$) with low rhythmicity in matched parasites that exhibit highly rhythmic expression in mismatched parasites were enriched to a single gene ontology

biological process term; RNA processing/splicing. This may represent the expression of an alternative set of IDC genes during rescheduling. In humans, a set of transcripts has been shown to gain rhythmicity in older individuals coinciding with the loss of canonical clock function[15].

Thirty-three percent of genes ($n = 1765$) fell under the threshold ($p > 0.05$) for rhythmicity in mismatched parasites. In matched parasites, a bimodal distribution of expression for these

**Fig. 1 *P. chabaudi* gene expression is sensitive to the phase of host circadian rhythms. a** Ring stage parasites from donor mice were used to infect recipient mice housed in two rooms that differed by 12 h in their light:dark cycle. Blood samples were collected for RNAseq analysis from day 4 post-infection every 3 h for 11 time points (N = 4 mice per group per time point). ZT is Zeitgeber time:hours after lights on. **b** Time series gene expression heatmap illustrates daily rhythmicity in matched parasites (left) that lost strong rhythmicity in mismatched parasites (right). Transcripts ordered in the same sequence based on their phase of expression. Each row represents a single gene, sorted according to the phase of maximum expression starting from first sample time point. The phase of expression of each gene was obtained from ARSER and N represents number of genes with 24 h rhythmicity in expression. Each time point is represented by expression heatmap of two biological replicates. Colors represent the row Z-score. **c** Venn diagram of the number of daily rhythmic genes in matched (top) and mismatched (bottom) parasites. **d** Transcripts with daily rhythmicity in both matched and mismatched parasites had lower median amplitude (0.86, brown dashed line) in mismatched parasites compared to matched parasites (1.22, blue dashed line). This was also the case for transcripts that had ~6 h delayed phase of expression in mismatched compared to matched parasites. **e** Histogram of the phase distributions of 1765 genes that displayed daily rhythmicity only in matched parasites. Solid black line indicates the mean circular phase, brown line represents the standard deviation of the mean of phases. Whilst these genes are not identified as having daily rhythms in mismatched parasites, their distribution is shown for comparison. **f** Transcripts that displayed daily rhythmicity only in matched parasites have median period close to 24 h (blue dashed line) and (for comparison) 23 h in mismatched parasites (brown dotted line). Source data are provided as Source Data file.

genes was observed with peaks at two different times of the day (ZT 8 and ZT 20, Fig. 1e) corresponding to the late trophozoite and ring stages of the IDC, respectively. Whilst these genes have a periodicity of expression extremely close to 24 h in matched parasites (median periodicity = 23.89 h), 55% of these transcripts had shorter periodicities in mismatched parasites (between 20 h to 24 h), which in turn reduced the overall periodicity by ~1 h (median periodicity 22.85 h) (Fig. 1f). Period estimates from genes that lost rhythms in expression profiles are used here to illustrate an overall trend, rather than provide information on individual genes. It is possible that the shorter periods observed for mismatched parasites correlates with a shorter IDC, explaining how mismatched parasites became rescheduled by approximately 6 h within four cycles of replication (an average of 1.5 h per cycle).

We undertook further analysis of the 1765 genes that fell under the threshold for rhythmicity of expression profile in mismatched parasites to examine which biological processes are affected. We divided genes into 12 groups based on the time of day (phase) of their maximal expression and performed gene ontology (GO)-based enrichment analysis within each group every 2 h (using the fitted model output from ARSER). The point when the highest relative copy numbers of mRNA from these genes was present was taken to be indicative of an overall trend. A wide range of biological processes including carbohydrate metabolism, nucleotide and amino acid metabolism, DNA replication, oxidation-reduction processes, translation, RNA transport, aminoacyl-tRNA biosynthesis, and ubiquitin-mediated proteolysis and proteasome pathways were enriched in different 2 h phase clusters, losing daily rhythmicity in host rhythm mismatched parasites (Fig. 2a and Supplementary Data 2). Many of these biological processes are under circadian clock control in other organisms[16,17]. Disruption of any of these processes could be detrimental to parasites and may explain the 50% reduction in parasite densities observed for mismatched parasites by O'Donnell et al.[7,8]. Next, we evaluate how these perturbed processes might relate to the IDC schedule.

Genes involved in energy metabolism pathways (glycolysis, fructose, and mannose metabolism) lost robust daily rhythmicity in mismatched parasites (Fig. 2a, b, Supplementary Data 2). Given that completing the IDC is glucose demanding, and host feeding rhythms affect the timing of the IDC, malaria parasites may be expected to express genes rhythmically to utilize this energy source efficiently[4,5]. Furthermore, the maximum relative number of transcripts of glycolysis-associated genes in matched *P. chabaudi* was observed between Zeitgeber (ZT) 22-2, which corresponds to the ring and early trophozoite stages of parasite development and also reflects the IDC stages when genes

associated with glycolysis are maximally transcribed in *P. falciparum* in vitro[12].

Nine out of 25 (36%) genes associated with the ubiquitin mediated proteolysis pathway, and 25 out of 32 (78%) genes encoding core and regulatory components of the proteasome system lost strong rhythmic expression in mismatched parasites (Fig. 2b, Supplementary Data 2). The ubiquitin-proteasome system (UPS) plays a direct role in determining the half-life of core clock components and other clock-controlled protein functions[18]. Genes that lost rhythmic expression include those associated with one E1 Ub-activating enzyme, four E2 Ub-conjugating enzymes, and three ring finger type E3 Ub-ligases (RBX1, SYVN, and Apc11) and the adapter protein SKP1 from the proteasome pathway. Some of these genes are essential for the yeast anaphase promoting complex (APC) which is a cell cycle regulated ubiquitin protein ligase[19–22]. The majority of UPS genes have a peak of expression between ZT 7.5–10 in matched parasites, which corresponds to the late trophozoite/schizont stage of the IDC (Fig. 2b). During the late IDC stage, the UPS helps parasites to shift from generic metabolic and cellular machinery to specialized functions.

Circadian clocks control the timing of DNA replication in many organisms[23–27]. We observed that 20 out of 43 (44%) genes associated with DNA replication lost strong rhythmic expression in mismatched parasites (Fig. 2a, b and Supplementary Data 2). This included genes encoding subunits of DNA polymerase, replication-licensing factors, DNA helicase and the DNA repair protein RAD51. Other cell-cycle associated genes encoding cdc2-related protein kinase 4 and 5, anaphase-promoting complex 4, cyclin 1, cullin like protein, regulator of initiation factor 2 and replication termination factor also lost rhythmic expression in mismatched parasites. Genes associated with DNA replication reached peak expression between ZT 8–12 in matched parasites, which corresponds to the transition from the late trophozoite to the schizont stage, and is when DNA replication machinery components are transcribed[12].

Seven out of 31 genes associated with cell redox and glutathione metabolism lost rhythmic expression in mismatched parasites and are enriched for the term "cell redox homeostasis" (Padj < 0.001; Supplementary Data 2). All seven genes displayed maximum expression during ZT 6–8 in matched parasites. Furthermore, peroxiredoxin proteins show circadian rhythmicity in oxidation/reduction cycles that are conserved across the tree of life[28]. Two out of three genes encoding peroxiredoxin showed daily rhythmic expression in both matched and mismatched parasites, while one gene (PCHAS_0511500) lost rhythmic expression in mismatched parasites. The expression of genes involved in redox metabolism is driven by an endogenous clock

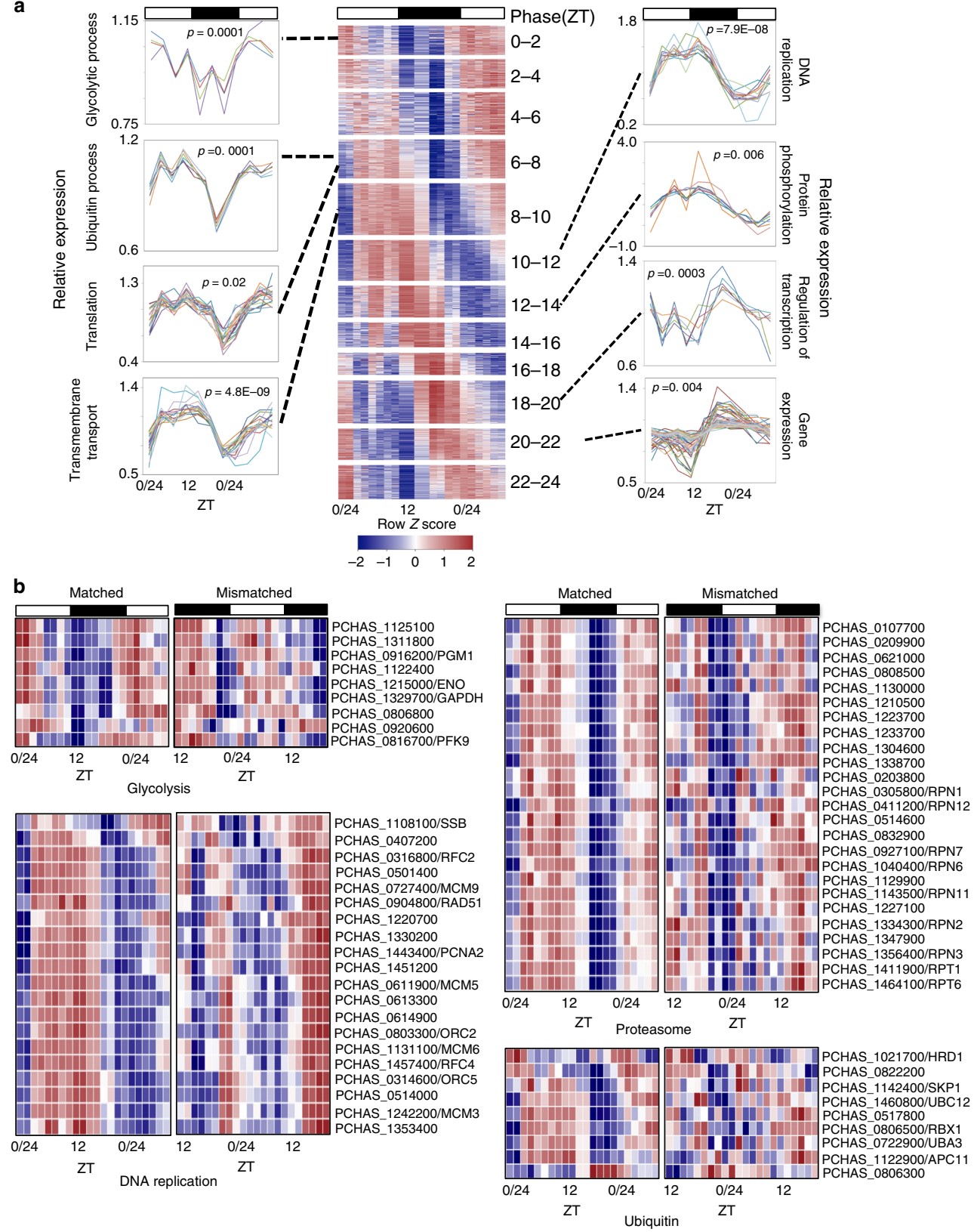

in *T. brucei*[3], which may be linked to rhythmicity in oxidation/reduction cycles resulting from host metabolic rhythms.

**Rhythms in the transcriptome of *P. falciparum*.** Parasites may set the timing of IDC transitions using a circadian clock that is entrained by a host rhythm (i.e., a Zeitgeber). One of the criteria for demonstrating clock control is that the rhythm persists (free-runs) under constant conditions. In vitro culture can provide constant conditions and in contrast to *P. chabaudi*, *P. falciparum* has an IDC of approx. Forty-eight hours, allowing putative 24 h clock-related genes and their downstream interactors to be

**Fig. 2 Biological pathways affected by mismatch to the phase of host rhythms. a** Time series gene expression view of genes that displayed with 24 h rhythmicity in matched parasites but lost rhythmicity in mismatched parasites. Genes were sorted based on phase of maximum expression and segregated into 12 groups with each group representing 2 h phase clusters. Line plots along the sides of the heat map represent expression profiles of individual genes significantly enriched to gene ontology terms (FDR corrected $p < 0.05$, hypergeometric test, one-sided) representing few crucial biological processes. The Y axis represents relative expression of genes at each time point determined by count level expression of each gene normalized by its mean across 11 time points. **b** Heatmaps illustrating the expression patterns of 24 h rhythmic genes for matched and mismatched parasites that are involved in the ubiquitin and proteasome systems, and the DNA replication and glycolysis pathways. These genes lost rhythmicity in mismatched parasites. Genes have been sorted based on the phase of maximum expression. The color scheme represents the row Z-score. Each time point is represented by the expression heatmap of two biological replicates.

distinguished from IDC genes. Observing a free-running rhythm of 24 h is consistent with the presence of a circadian clock, but other factors such as temperature compensation and entrainment must also be demonstrated to conclusively prove the presence of a clock.

To determine whether *P. falciparum* possesses a 24 h free-running transcriptome, we carried out time-series RNAseq experiments to observe the expression profile of all *P. falciparum* genes at a 2 h resolution from highly synchronized parasites grown at a constant temperature and in constant darkness (see Methods). RNAseq data from two biological replicates per time-point were obtained. We identified 361 transcripts (~6% of the total genes) with 24 h rhythmicity (common rhythmic genes detected by ARSER and JTK cycle with $q < 0.05$) from the expression profiles of 5702 genes (Fig. 3a, Supplementary Data 3). The median amplitude of oscillations of the 24 h free-running genes was 0.93. This is lower than the amplitude of the rhythmic genes detected in *P. chabaudi* in vivo (1.22). A higher amplitude in *P. chabaudi* may be due to reinforcement by host rhythms and/or the contribution of genes expressed due simply to 24 h IDC developmental progression—neither of which apply to *P. falciparum* in vitro. Genes identified in *P. falciparum* with a 24 h expression pattern are enriched for most of the processes that exhibit reduced rhythmicity in mismatched *P. chabaudi* infections (Ubiquitin proteasome system, regulation of cell cycle, nuclear division, protein localization, and transmembrane transport) (Fig. 3b) but not redox processes. This suggests that circadian rhythms in the redox state of RBCs[29], which persist when RBCs are cultured in constant conditions, are not important for the timing of developing parasites.

Comparing the expression profiles of orthologues of *P. falciparum* that were rhythmic in matched *P. chabaudi* (333 one to one orthologues identified) but lost rhythmicity in mismatched *P. chabaudi* ($N = 1765$) identified 103 common daily rhythmic genes. Taken together, these observations suggest that a set of genes whose expression is sensitive to the timing of host daily rhythms and which are also rhythmic under constant conditions are potentially driven by an endogenous oscillator.

**Role of serpentine receptor 10 in the IDC**. If malaria parasites are able to schedule the IDC they must respond to time-of-day information either through receptors or transporters. Seven transmembrane domain-containing receptors/serpentine receptors/G protein-coupled receptors (GPCRs) are the largest and most diverse group of membrane receptors and participate in a variety of physiological functions[30–33].

The *P. falciparum* proteome contains four serpentine receptor (SR) proteins; SR1, SR10, SR12, and SR25[34]. Of these, only *Pfsr10* showed a 24 h expression rhythm (Fig. 3c) while the rest showed periodicity closer to 48 h (Supplementary Fig. 3a). Serpentine receptor 10 (*sr10*: PF3D7_1215900), was also the top ranked receptor in the 24 h rhythmic *P. falciparum* gene list (ranked 28 out of all 361 genes sorted based on *q*-values from JTK output) (Supplementary Data 3). Additionally, SR10 has been

bioinformatically classified as a member of Class A serpentine receptors belonging to the hormonal receptor subclass based on the length of the N-terminal domain[34] and classification by Inoue et al.[35]. Its orthologue in *P. chabaudi* (PCHAS_1433600) also has a 24 h expression rhythm in both matched and mismatched parasites (Fig. 3c, Supplementary Data 1). In *P. falciparum*, peak *sr10* expression corresponds to the ring (8 h post-invasion) and late trophozoite stages (32 h) of the IDC. In *P. chabaudi*, expression peaked at ZT 14, corresponding to the late trophozoite stage. SR10 is also the only receptor shared between malaria parasites and other apicomplexans and also with distantly related organisms such as *Caenorhabditis elegans*, *Drosophila melanogaster*, *Gallus gallus*, *Mus musculus*, *Homo sapiens*, and *Arabidopsis thaliana* (data retrieved from OrthoMCL DB), although it has not been linked to circadian clocks in these organisms.

**Serpentine receptor 10 influences IDC duration**. To investigate whether SR10 may be part of a receptor-mediated signaling system in malaria parasites that influences the IDC based on time-of-day information from the host, we disrupted the *sr10* gene in *P. chabaudi* by a double crossover homologous recombination strategy to generate *sr10* deficient parasite clones (*sr10*KO) (Supplementary Fig. 3b). Functional disruption of *sr10* was verified by RNAseq analysis (Supplementary Fig. 3c). We compared the IDC schedule of wild type and two *sr10*KO clones by examination of thin blood smears ($n = 4$ per group), every three hours over 48 h, starting from day 1 PI at ZT 13.5 (Fig. 3d). All infections of wild type and *sr10*KO clones (*sr10*KOA and B) were highly synchronous (amplitude ± SE) for *P. chabaudi* wild type: 0.94 ± 0.02, *P. chabaudi sr10*KOA: 0.79 ± 0.02 and *sr10*KOB 0.93 ± 0.03, Fig. 3e, Supplementary Table 1). Period estimates for the proportion of parasites at early trophozoite stage suggest the IDC duration of both *sr10*KO clones is ~2–3 h shorter (IDC duration 22.4 h) than the wild type (IDC duration 25.15; $p < 0.0001$, Fig. 3f). We repeated this using the same strategy for *P. yoelii* (Supplementary Fig. 3d). The proportion of parasites at early trophozoite stage displayed weak daily rhythmicity in both the wild type and *sr10*KO clone (amplitude for *P. yoelii* wild type: 0.30 ± 0.02 and *P. yoelii sr10*KO: 0.31 ± 0.01, Fig. 3g, Supplementary Table 1), yet the IDC duration was shorter in the *sr10*KO clone (*sr10*KO IDC duration 24.45 h; wild type IDC duration ~28 h, Fig. 3h). Observing similar changes to IDC duration in two different experiments using two different malaria species strongly implicates *sr10* in the control of developmental progression through the IDC.

**SR10 regulates expression of multiple pathways**. To explore how disruption of *sr10* affects IDC progression we repeated the time-series RNAseq experiments on both wild type and *sr10*KO *P. chabaudi* parasites from 17 time points (however, data from the first three time points were excluded owing to a low number of mapped reads i.e., <1 million paired ends mapped reads) sampled every 3 h ($n = 2$ per time point), starting from day 2 PI

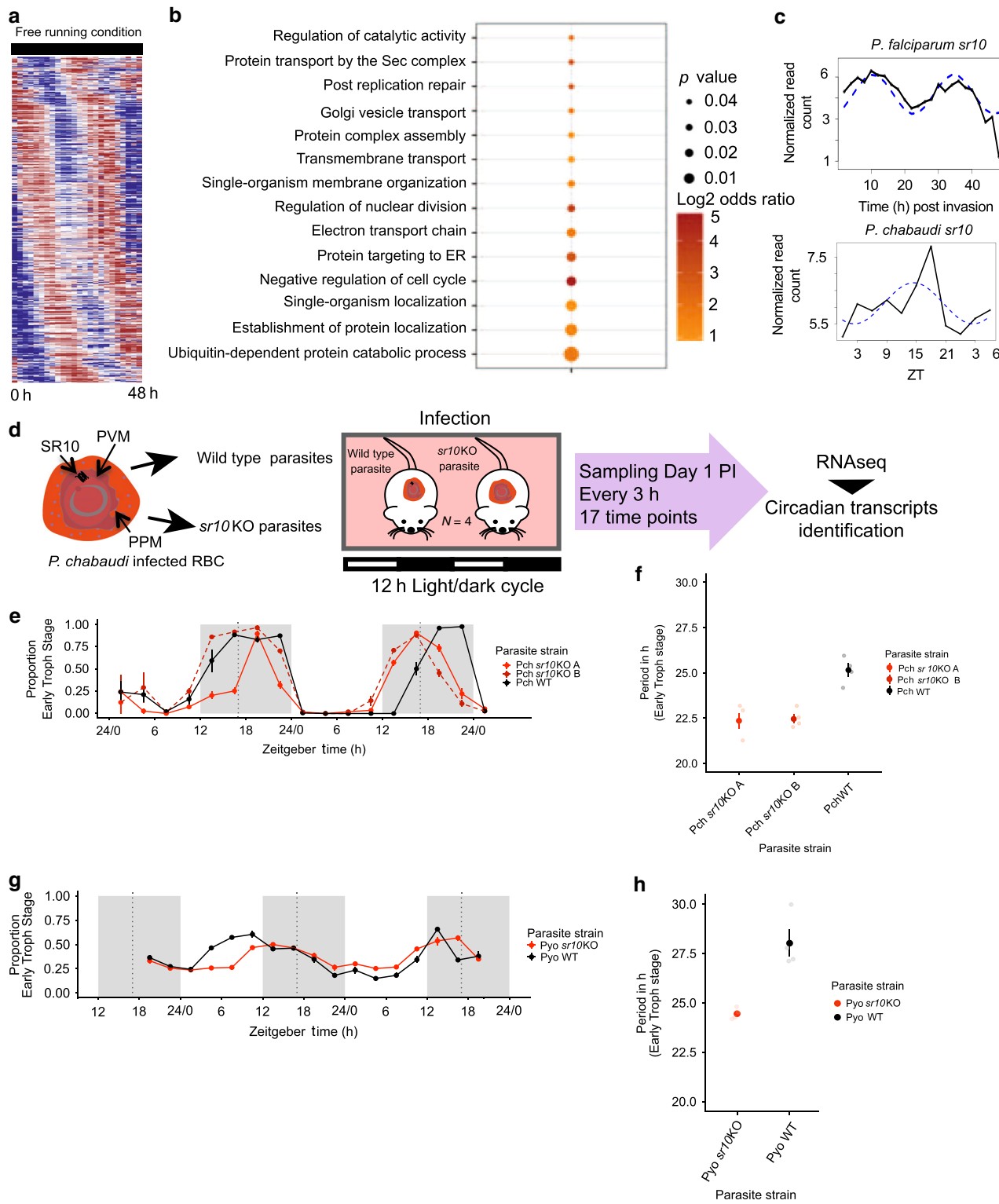

(05.30 h, ZT 22.5). Biological replicates were found to be tightly clustered (Supplementary Fig. 3e). Principal component analysis of 14 time points identified that the first and third components of PCA (with a cyclic pattern in both the wild type and *sr10*KO parasites), accounted for >85% of total variance (Supplementary Fig. 3f). A total of 3620 and 2886 genes showed ~ 24 h rhythmicity in expression in *P. chabaudi* wild type and *sr10*KO parasites respectively ($q < 0.05$; Supplementary Fig. 4, Supplementary Data 4). 1015 genes (19% of the total genes) had dampened rhythms in expression in *sr10*KO parasites (Fig. 4a, b,

Supplementary Data 4). Furthermore, 85% of the genes identified as being transcribed with daily rhythms in matched *P. chabaudi* parasites from our first experiment also had a daily rhythm in wild-type parasites in this dataset (which are also matched). Whilst the additional rhythmically expressed genes identified in wild type parasites in this dataset could be due to a longer time series, we found generally high concordance between the transcriptomes of infections initiated in the same way but in different laboratories. This lends support to the inference that the genes losing rhythmicity of expression (henceforth called SR10-linked

**Fig. 3 Serpentine receptor 10 maintains the duration of the intra-erythrocytic developmental cycle. a** Time series RNAseq gene expression heatmap view of 24 h rhythmic genes identified in *P. falciparum* in vitro in "free running" (constant temperature and darkness) conditions. Genes sorted based on the phase of maximum expression starting from time T:0 that corresponds to 3 h post merozoite invasion. **b** Manually curated gene ontology terms enriched for *P. falciparum* genes with 24 h expression (FDR corrected $p < 0.05$, hypergeometric test, one-sided). **c** Line graphs represent the expression of serpentine receptor 10 in *P. falciparum* over its 48 h IDC (top plot) and in *P. chabaudi* during its 24 h IDC (bottom plot). Dotted lines show the best-fit sinusoidal curves. **d** *P. chabaudi* wild type and sr10KO parasites were used to initiate infections in CBS mice. Blood was collected from day 1 (ZT 13.5) every 3 h during the following 48 h. Expression data from two biological replicates over 14 time points (from day 2 PI, ZT 22.5) were analyzed to identify putative "circadian" transcripts. PVM parasitophorous vacuole membrane, PPM parasite plasma membrane, RBC red blood cell. **e** Proportion of parasites in the blood at early trophozoite stage in *P. chabaudi* wild type (WT) and sr10KO clones (Mean ± SEM, $N = 4$/clone). **f** IDC duration of *P. chabaudi* wild type (WT) and sr10KO clones (Mean ± SEM, $N = 4$/clone). **g** Proportion of parasites in the blood at early trophozoite stage in *P. yoelii* wild type (WT) and sr10KO clones (Mean ± SEM, $N = 4$/clone). **h** IDC duration of *P. yoelii* wild type (WT) and sr10KO clones (Mean ± SEM, $N = 4$/clone). Source data are provided as Source Data file.

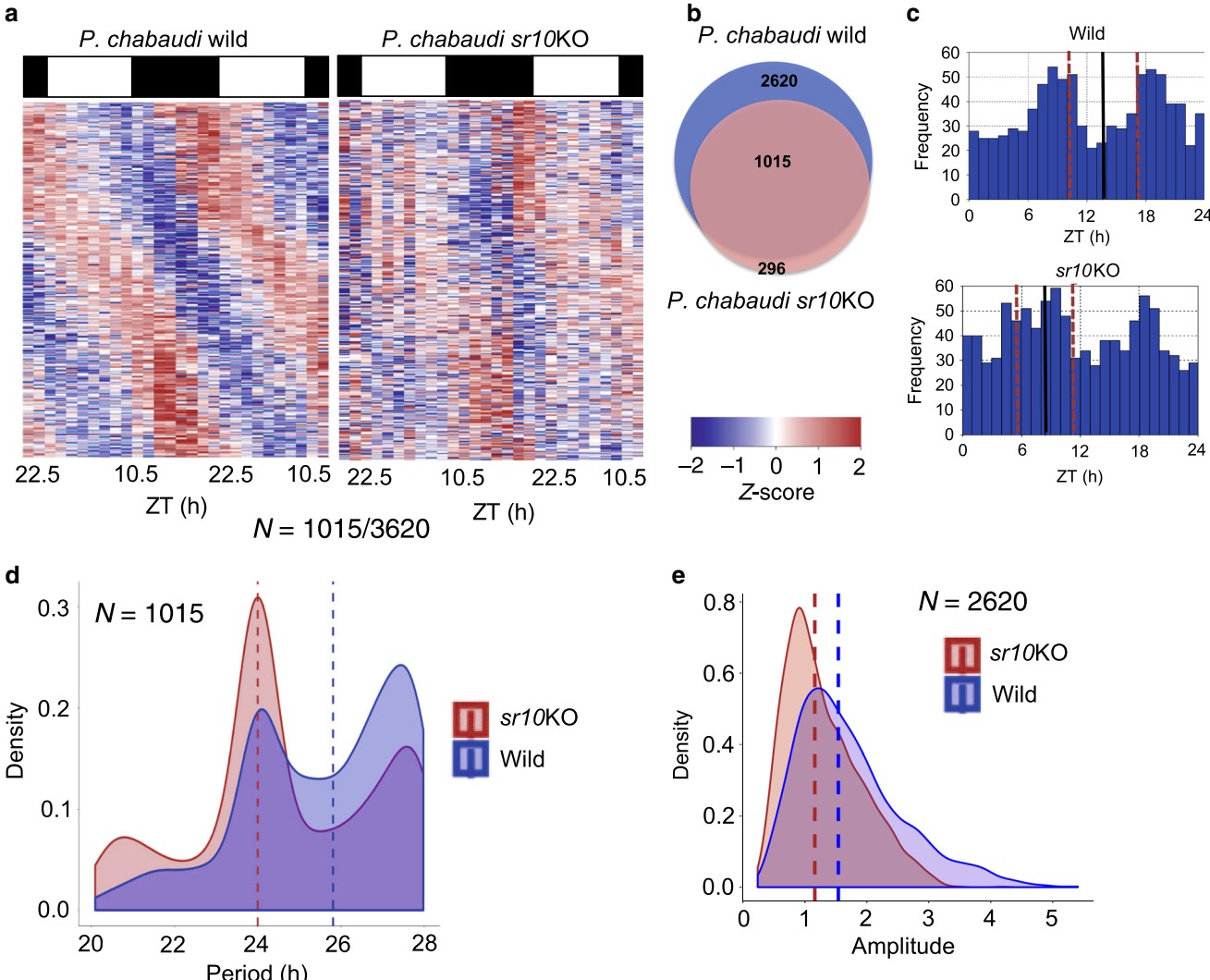

**Fig. 4 Disruption of *sr10* affects daily rhythms in *P. chabaudi* gene expression. a** Time series gene expression heatmap view of genes expressed with daily rhythmicity in *P. chabaudi* wild type that lost rhythmicity in *sr10*KO parasites. Right most heatmap shows the expression pattern of transcripts that lost rhythmicity in *sr10*KO parasites. Each row represents a single gene, sorted according to phase of maximum expression starting from first time point of sample collection. *N* represents number of genes with 24 h rhythmicity identified. Each time point is represented by an expression heatmap of two biological replicates. **b** Venn diagram of number of genes with 24 h rhythmic expression identified by both JTK and ARSER in wild type and *sr10*KO parasites. **c** Phase distributions of genes with 24 h expression in wild type that lost 24 h rhythmicity in *sr10*KO parasites. The mean circular phase for each condition is indicated by a solid black line. *N* represents the number of cycling transcripts. Pink lines represent standard deviation of the mean circular phases. **d** Transcripts that displayed with 24 h (putative "circadian") rhythmicity only in wild type parasites have median periods close to 26 h in wild-type parasites (blue dashed line) and 24 h in *sr10*KO parasites (brown dashed line). **e** Genes that were rhythmic in both wild type and *sr10*KO parasites had a significantly lower mean amplitude in *sr10*KO parasites (1.15, brown dashed line) compared to wild type parasites (1.53, $p < 0.00001$). Source data are provided as Source Data file.

rhythmic genes or "SLRGs") in *sr10*KO parasites is due to the loss of *sr10*.

Examination of SLRGs revealed a bimodal distribution pattern for peak expression in wild type parasites (peaking at ZT 8 and ZT 19). This pattern was partially lost in *sr10*KO parasites in which the early peak displays a broader distribution (Fig. 4c). Further, the SLRGs exhibit a shorter periodicity in *sr10*KO (24 h) compared to wild type (25.81 h) parasites (Fig. 4d). Our intention is not to draw inference from the quantitative difference of periods (which is of limited utility for genes that lose rhythmicity), but rather to ascertain a qualitative comparison. Nonetheless, the shorter periods in *sr10*KO parasites reflects the shorter periods observed in genes that lose rhythmicity in mismatched parasites in our first experiment. As for genes that retained rhythmic expression in both matched and mismatched parasites, the genes that retained rhythmic expression in only *sr10*KO parasites ($N = 2620$) exhibited a significant reduction ($p < 0.0001$, unpaired Student's $t$-test) of amplitude in *sr10*KO (1.15) compared to wild type parasites (1.53) (Fig. 4e).

GO analysis of the SLRGs revealed enrichment for terms related to translation, RNA splicing, RNA and vesicle-mediated transport and purine and pyrimidine metabolism, indicating a broad effect of SR10 loss on parasite biology (Supplementary Fig. 5, Supplementary Data 5). Comparing differentially regulated genes for four different IDC stages (i.e., four time points: Day 2 ZT 16.5, Day 2 ZT 22.5, Day 3 ZT 4.5, and Day 3 ZT 10.5) in wild-type and *sr10*KO parasites (Fig. 5a, Supplementary Data 6) revealed that (i) genes associated with protein translation are perturbed in ring stages; (ii) DNA replication and cell cycle associated processes are perturbed in early and late trophozoite stages; and (iii) microtubule-based movements are perturbed in schizonts (Fig. 5b). We then compared the transcripts that lost rhythmicity in mismatched parasites ($N = 1765$) to the SLRGs ($N = 1015$). A total of 326 genes were shared (Supplementary Fig. 6a) suggesting that their expression is shaped by both host rhythms and how the IDC is scheduled by the parasite's expression of *sr10*. The shared genes were enriched for biological processes including energy metabolism, heme metabolic processes, and translation (Supplementary Fig. 6b).

**SR10 affects rhythmic expression of spliceosome machinery.** The spliceosome is a large and dynamic ribonucleoprotein complex of five small nuclear ribonucleoproteins (snRNP) and over 150 multiple additional proteins that catalyze splicing of precursor mRNA in eukaryotes[36,37]. Out of 85 genes (based on the Kyoto Encyclopedia of Genes and Genomes (KEGG) database mapping) expressing different spliceosomal proteins in *P. chabaudi*, 44 genes showed 24 h expression rhythms in wild type parasites, of which 26 ($q < 0.005$, hypergeometric test) are in the SLRG group (Fig. 6a). They represent proteins of major spliceosome components including core spliceosomal protein members of snRNPs, prp19 complex and prp 19 related complexes. Alternative splicing can regulate gene expression in signal dependent and tissue-specific manners[38] and an emerging body of evidence links alternative splicing with the control of circadian regulatory networks in a variety of organisms, including *Drosophila melanogaster*[39], *Neurospora crassa*[40,41], *Arabidopsis*[42,43], and *Mus musculus*[44]. Alternative splicing has been reported to occur in Apicomplexans (including malaria parasites) for relatively few genes, covering only several percent of the total genes[45].

Having observed that the loss of *sr10* modulates the expression of genes associated with the spliceosome, we investigated whether it also affects the alternative-splicing (AS) signature using RNAseq analysis of both wild type and *sr10*KO parasites. Comparison of *sr10*KO and wild-type parasites identified 320 differential alternative splicing events covering 214 genes ($p < 0.05$) for ZT 22.5 and 708 differential alternative splicing events covering 409 genes ($p < 0.05$) for ZT 1.5 (Fig. 6b, Supplementary Data 7) (for details see Methods section). In a separate analysis (see Methods), we found only 72 (54 genes) and 151 (114 genes) differential alternative splicing events in each strain when we compared two consecutive time points (i.e., ZT 22.5 vs. ZT 1.5) (Fig. 6b, Supplementary Data 7). This suggests that SR10 impacts the spliceosome machinery, resulting in differential alternative splicing patterns. GO-enrichment analysis of genes that showed differential alternative splicing events enriched with biological process terms such as translation, intracellular signal transduction and protein sumoylation ($P$adj < 0.05) of *sr10*KO compared to wild-type parasites. These observations collectively suggest that SR10 may link host derived time-of-day information with the IDC schedule and regulate alternative splicing.

**High-throughput real-time qPCR based validation.** We independently verified the expression patterns of 87 genes that lost rhythmicity either in the mismatched parasites or in *sr10*KO parasites through high-throughput real-time qPCR using the BioMark[TM] HD system (Fluidigm). A total of 58 genes, from the initial experiment comparing matched and mismatched *P. chabaudi* parasites, covering 11 randomly selected genes that represent multiple affected pathways were tested for their expression in both groups (Supplementary Fig. 6c). Similarly, a total of 36 genes from the *sr10*KO experiment covering 12 randomly selected genes and representing multiple affected pathways were tested for their expression in both *P. chabaudi* wild type and *sr10*KO strains (Supplementary Fig. 6d). The gene datasets, generated by high-throughput real-time qRT-PCR using the BioMarkHD platform tightly correlated with the RNAseq expression values (Spearman-rank correlation between 0.67 −0.95 with $p < 0.05$ for all the genes tested), thus independently validating our RNAseq analysis.

To validate the accuracy of *P. falciparum* time-series RNAseq datasets, the expression profiles of nine randomly chosen genes were validated using real-time qPCR; expression patterns were tightly correlated (Supplementary Fig 6e).

## Discussion

How malaria parasites interact with host rhythms to establish and maintain a schedule for IDC development is unknown. Our analyses, which were carried out using the rodent malaria parasites *P. chabaudi* and *P. yoelii* in vivo, and the human malaria parasite, *P. falciparum* in vitro, reveal an extensive transcriptome with 24 h rhythmicity and suggest that coordination of the IDC with host rhythms is important for the parasites' ability to undertake key cellular processes. This includes metabolic pathways, DNA replication, redox balance, the ubiquitin proteasome system, and alternative splicing (Fig. 6c). Twenty-four hour rhythmicity in almost all of these processes persists in conditions in which parasites are not exposed to host rhythms, suggesting the presence of an endogenous time-keeping mechanism. We also found that the IDC duration is, at least in part, controlled by serpentine receptor 10 (SR10). Given its role in determining the duration of IDC and being a GPCR class of receptor, we propose that SR10 acts as a link between host circadian rhythms and the parasite's endogenous time-keeping/IDC scheduling mechanism. In support of this hypothesis, SR10: (i) has a 24 h rhythmic expression in *P. falciparum*, and in both matched and mismatched *P. chabaudi*; (ii) regulates rhythmicity in gene expression for several of the processes whose genes lost rhythmic expression in mismatched *P. chabaudi* (Fig. 6c); and (iii) determines the duration (period) of gene expression patterns when it is disrupted.

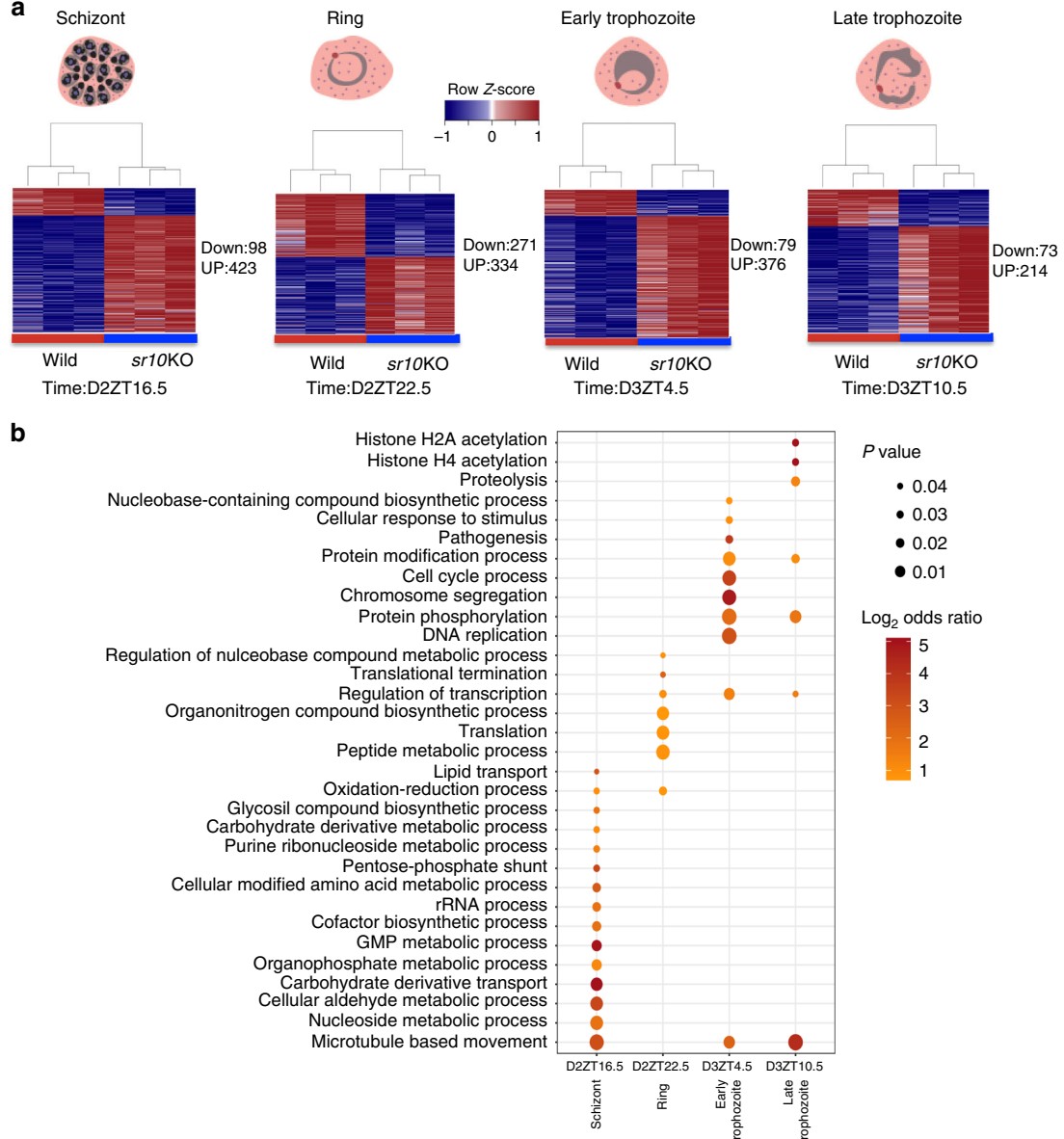

**Fig. 5 Knock out of *sr10* affects multiple biological processes. a** Differentially regulated genes were identified by comparing four matching time points of *sr10*KO and wild type *Plasmodium chabaudi* parasites. Up and down represent differentially regulated genes with the false discovery rate corrected $p < 0.05$ and $Log_2$ fold change $< -1$ for downregulated genes and $> 1$ for upregulated genes at each time point. The four time points analyzed represent four IDC stages as derived from examination of parasite morphology in thin blood smears. **b** Gene ontology analysis of the differentially regulated genes within each time point. Manually curated functional annotations of biological processes (False discovery rate corrected $p < 0.05$, hypergeometric test, one-sided) are represented and the color spectrum represents the odds ratio.

Most genes (3057 of 5343) in the transcriptome of *P. chabaudi* in synchrony (matched) with the host circadian rhythm are transcribed with 24 h periodicities, whilst this number drops to 1824 when parasites are mismatched to the host circadian rhythm. Genes that lose rhythmicity are involved in diverse biological processes including glycolytic process, DNA replication, translation, the ubiquitin/proteasome pathway and redox metabolism (Supplementary Data 2). This is not simply a consequence of mismatched parasites becoming desynchronized because they maintain morphological synchrony during rescheduling (Supplementary Fig. 1a). Instead, disruption to these processes could be a result of stresses resulting from the IDC being misaligned to host rhythms. If, for example, rescheduling parasites are unable to coincide the appearance of a particular IDC stage with a rhythmically provided resource it needs from the host[4–6], the parasite may be physiologically compromised and alternative pathways upregulated. Such stresses may explain the reduced parasite densities previously observed in mismatched *P. chabaudi*[7,8]. Whilst caution needs to be employed in interpreting periodicities from short time-series and from genes that exhibit dampened rhythmicity, on the whole, the expression patterns of such genes were approximately 1 h shorter in mismatched than matched parasites. This difference coincides with the observation that mismatched parasites were rescheduling by on average approximately 1.5 h every IDC.

The IDC of *P. chabaudi* is approximately 24 h in duration, making it difficult to distinguish genes associated with features of particular IDC stages from genes associated with a time-keeping mechanism or its outputs. Thus, *P. falciparum* cultured under constant conditions was used to separate IDC genes from putative

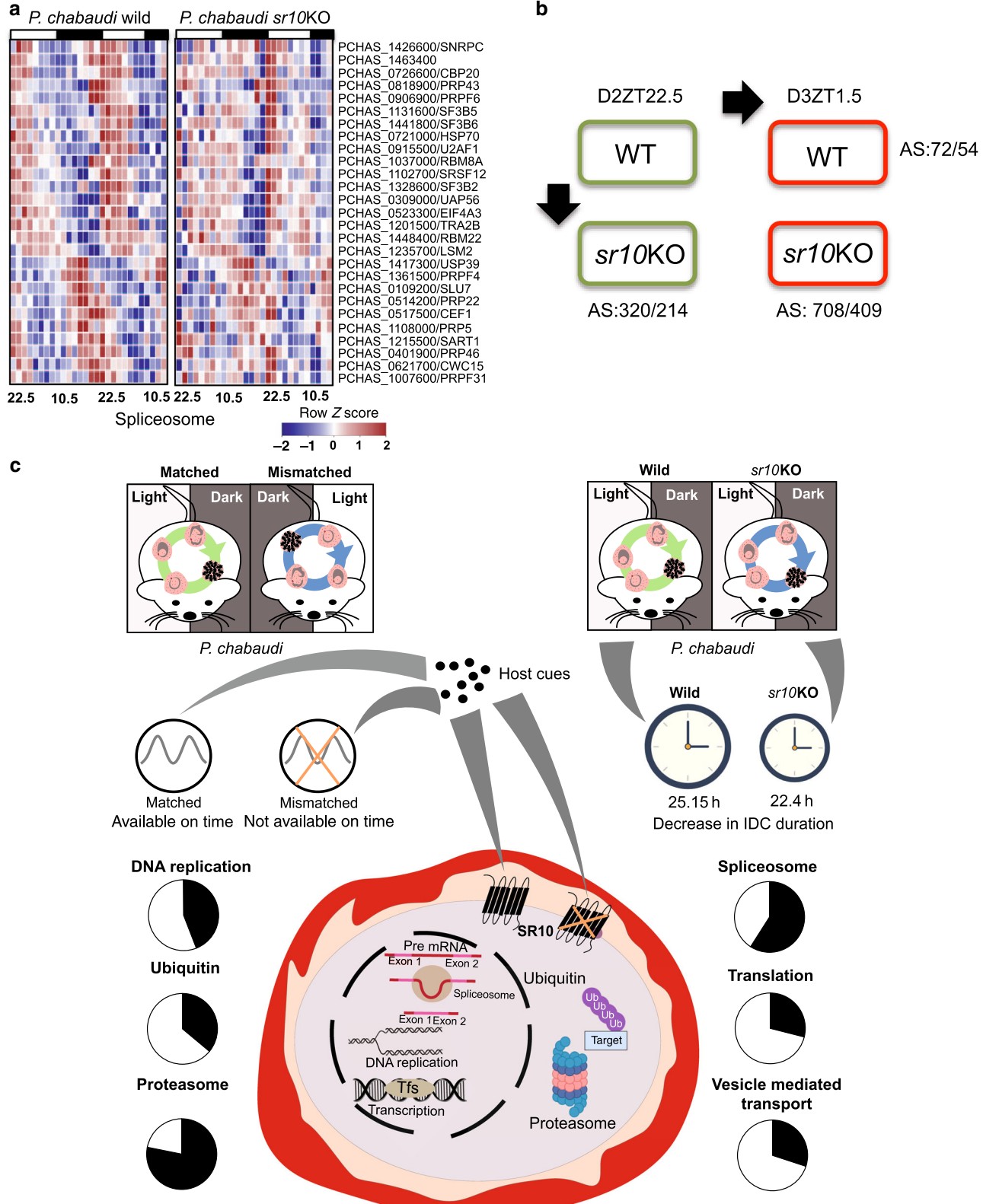

time-keeping associated genes. We found that 361 *P. falciparum* genes exhibit ~24 h rhythmic expression and many of these genes are associated with processes (regulation of cell cycle, nuclear division, transmembrane transport, and the ubiquitin proteasome system) affected in *P. chabaudi* by mismatch to host rhythms. Finding that the ubiquitin–proteasome system is disrupted in mismatched parasites is of particular interest because this system

plays a role in regulating clock components and their outputs in many taxa[18].

We also observed rhythmic expression of genes associated with histone modification and the control of transcription and translation (Figs. 2a, 3a, 5b and Supplementary Fig. 3). These processes are considered central to the circadian organization of the transcriptome in other taxa[46–50].

**Fig. 6 Cross-talk between the intraerythrocytic developmental cycle and host rhythms. a** *sr10* knockout affects parasite spliceosome machinery. Heatmap illustrating the expression pattern of *sr10* knockout affected 24 h rhythmic genes involved in spliceosome pathway in *P. chabaudi* wild type and *sr10*KO parasites. The list of genes was obtained by mapping the SR10 linked rhythmic genes (SLRGs) to *P. chabaudi* spliceosome pathway represented in the KEGG database. Genes have been sorted based on phase of maximum expression. The color scheme represents the row Z-score. **b** *sr10* knockout affects alternative splicing signature of the transcriptome. *sr10* knock out affects the alternative splicing signature of the parasite transcriptome. Two consecutive time points were compared between wild type and *sr10*KO parasites to identify differential usage of exons. As a control two consecutive time points (Day 2 ZT 22.5 and Day 3 ZT 1.5) from the same parasite strain were also compared. The number shown depicts the number of differential exon usage events detected ($p < 0.05$). Two biological replicates per time point were used. Differential exon usage events were identified using DEXSeq[68]. **c** Schematic figure summarizing the cross-talk between the IDC schedule of *P. chabaudi* and host rhythms. The parasite can reschedule its IDC when its developmental rhythms are mismatched with the host rhythms. The parasite responds to mismatch by losing rhythmic expression of genes associated with multiple biological processes as depicted in the pie-charts on the left. *P. chabaudi* serpentine receptor 10 (*sr10*) is expressed rhythmically during the IDC, and knocking out *sr10* in *P. chabaudi* reduces the IDC duration by ~2–3 h and also affects the rhythmic expression of genes associated with multiple biological processes as depicted in the pie charts on the right. We speculate that SR10 may serve as one of the receptors through which the parasite receives rhythmic cues from the host that influence the IDC schedule, permitting rescheduling to recover from mismatch. Black section within the pie-charts represent the percentage of rhythmic genes in each biological process that fell under the threshold for rhythmic expression in mismatched and *sr10*KO parasites.

In summary, we reveal that coordination with host daily rhythms is central to the schedule of expression of genes associated with diverse processes underpinning IDC progression, and ultimately, within-host replication. Our data are consistent with some of the criteria required to demonstrate an endogenous time-keeping ability (free-running rhythm), and further work is required to examine whether other features of an endogenous clock exist, as well as to identify the interaction partners that link SR10 with host time-of-day. The IDC underpins malaria parasites' capacity to undergo rapid asexual replication and cause severe disease, and fuels transmission of the disease. Thus, uncovering the components of the parasites' time-keeping mechanisms and the signaling system that links it to IDC progression may reveal novel intervention strategies.

## Methods

**Ethics statement**. All animal procedures for the parasite mismatch study were performed in accordance with the UK Home office regulations (Animals Scientific Procedures ACT 1986; project license number 70/8546) and approved by the University of Edinburgh. All the procedures for the *sr10*KO studies were performed in accordance with the Japanese Humane Treatment and Management of Animal Law (Law No. 105 dated 19 October 1973 modified on 2 June 2006), and the Regulation on Animal Experimentation at Nagasaki University, Japan. The protocol was approved by the Institutional Animal Research Committee of Nagasaki University (permit: 12072610052). The animal experiments conducted in Edinburgh and Nagasaki were officially exempted from additional IBEC clearances in KAUST. All the procedures to perform work on different parasite materials used in this study were approved by IBEC in KAUST (IBEC number: 19IBEC12).

**Experimental design and data collection**. For the host rhythm mismatching experiment, hosts were 6–8 weeks old female MF1 mice housed in groups of five in open top cages at 21 °C and 65% humidity with food and drinking water supplemented with 0.05% para-aminobenzoic acid (PABA, to supplement parasite growth) provided ad libitum. All experimental infections were initiated by intravenous injection of $1 \times 10^7$ *P. chabaudi chabaudi* clone AS[51] parasitized red blood cells (per ml). Parasites at ring stage were collected at ZT 0.5 (08:00 GMT) from donor mice housed in standard LD light conditions (Lights ON 07:30; Lights OFF 19:30 GMT) and immediately used to infect two groups of experimental mice. One group (termed "matched") were entrained to the same photoperiod as the donor mice, generating infections in which parasite and host rhythms were in the same phase. The other group (termed "mismatched") was entrained to reverse light (Lights OFF 07:30 and Lights ON 19:30 GMT) resulting in infections in which the parasite is out of phase with the host by 12 h.

Samples were collected every 3 h, over a period of 30 h, from 09:00 GMT on day 4 to 15:00 GMT on day 5 post infection. This corresponds to a starting time of ZT 1.5 for matched, and ZT 13.5 for mismatched infections. At each sampling point, four mice from each group (matched/mismatched) were sacrificed and thin blood smears and RBC counts (via flow cytometry; Beckman Coulter) were taken by tail bleeds, and 50 μl of blood was taken via cardiac puncture (added to 200 μl of RNAlater and frozen at −80 °C) for RNAseq analysis. The developmental rhythm of parasites was assessed from blood smears, in which the number of parasites at each of three morphologically distinct stages (ring stage trophozoite (hereby referred to as "ring stage"), early (small) trophozoite stage, late (large) trophozoites,

and schizont; differentiated based on parasite size, the size and number of nuclei and the appearance of haemozoin) was recorded.

The *sr10* gene knockout and subsequent experiments were performed with the *P. chabaudi* AS clone. Routine maintenance of the parasites was performed in ICR female mice (6–8 weeks old) housed at 23 °C with 12 h light–dark cycle (lights-off: 19:00 h and lights-on: 07:00 h) and fed on a maintenance diet with 0.05% PABA-supplemented water. Rhythmicity was assessed in groups of female CBA inbred mice (6–8 weeks old). Mice (SLC Inc., Shizuoka, Japan) were housed at 23 °C with 12 h light–dark cycle (lights-off: 19:00 h and lights-on: 07:00 h) and fed on a maintenance diet with 0.05% PABA-supplemented water.

Ring stage *sr10*KO (clone A and B) and wild type *P. chabaudi* parasites were sub-inoculated into groups of four CBA mice each ($1 \times 10^6$ parasitized RBCs per mouse) by intravenous injection at 20:30 h corresponding to ZT 13.5, day 0. Starting at ZT 13.5 on day 1 post-infection, blood smears were taken for both groups every three hours for 48 h producing a total of 17 time points. Blood smears were briefly fixed with 100% methanol and stained with Giemsa's solution. IDC stages were recorded based on classification of the parasitic forms into four stages —rings (ring stage trophozoite), early (small) trophozoites, late (large) trophozoites and schizonts. The same procedures were adopted for *P. yoelii* wild type ($17 \times 1.1$ pp) and *sr10*KO clones to obtain time series phenotype data using blood smears. Blood microsamples were also collected at these timepoints for time-series gene expression analysis[52]. Briefly, 20 μl of blood was collected via tail snip at each time point, washed in PBS and immediately treated with 500 μl TRIzol reagent and stored shortly at 4 °C and for long term at −80 °C.

***Plasmodium falciparum* culture**. *Plasmodium falciparum* II3 strain[53] (a DiCre recombinase expressing parasite derived from 3D7 clone) was maintained at 37 °C, 5% $O_2$, and 5% $CO_2$ in AB+ human RBCs in RPMI 1640 medium containing Albumax II (Invitrogen) supplemented with 2 mM L-glutamine. Culture medium was changed in every two days to avoid providing unknown 24 h rhythmic cues that might be present in the culture medium. Cultures were routinely monitored by Giemsa's solution-stained thin blood smears and synchronized by treating the cultures with 5% sorbitol (w/v) to kill any late stage asexual stages. For the time-series experiment, mature segmented schizonts were isolated by centrifugation over cushions of 70% (v/v) isotonic Percoll (GE healthcare Life Sciences), washed twice with RPMI 1640 without Albumax and allowed to invade fresh RBCs for about 2 h in a shaker maintained at 37 °C. Following invasion, the remaining schizonts were removed from the culture by overlaying the culture over cushions of 70% (v/v) isotonic Precool (GE healthcare Life Sciences). The pellet containing ~2 h post-invasion early ring stage infected RBCs and uninfected RBCs was then treated with 5% sorbitol (w/v) to kill contaminating mature schizonts. The culture containing highly synchronous early ring stage parasites was then split into six-well plates. Over the course of 48 h, 1–2 ml of infected RBC culture was collected every 2 h, media was removed by centrifugation for 2 min at $900 \times g$, and cells were lysed by adding 1 ml of TRIzol and immediately stored at −80 °C. The T0 time point was considered to be 3 h after the addition of Percoll cushion isolated segmented mature schizonts to the fresh RBCs.

**Time series gene expression analysis using RNAseq**. Total RNA was isolated from TRIzol treated samples according to the manufacturer's instructions (Life Technologies). Strand-specific mRNA libraries were prepared from total RNA using TruSeq Stranded mRNA Sample Prep Kit LS (Illumina) according to the manufacturer's instructions. Briefly, at least 100 ng of total RNA was used as starting material to prepare the libraries. PolyA+ mRNA molecules were purified from total RNA using oligo-T attached magnetic beads. First strand synthesis was performed using random primers followed by second strand synthesis where dUTP were incorporated in place of dTTP to achieve strand-specificity. Ends of the

double stranded cDNA molecules were adapter ligated and the libraries amplified by PCR for 15 cycles. Libraries were sequenced on Illumina HiSeq 4000 platform with paired-end 100/150 bp read chemistry according to manufacturer's instructions.

RNAseq read quality was assessed using FASTQC quality control tool (http://www.bioinformatics.babraham.ac.uk/projects/fastqc). Read trimming tool Trimmomatic[54] was used to remove low quality reads and Illumina adapter sequences. Reads smaller than 36 nucleotides long were discarded. Quality trimmed reads were mapped to *P. chabaudi chabaudi* AS reference genome (release 28 in PlasmoDB—http://www.plasmodb.org) using TopHat2 (version 2.0.13)[55] with parameters "–no-novel-juncs –library-type fr-firststrand". Gene expression estimates were made as raw read counts using the Python script "HTSeq- count" (model type—union, http://www-huber.embl.de/users/anders/HTSeq/)[56]. Count data were converted to counts per million (cpm) and genes were filtered if they failed to achieve a cpm value of 1 in at least 30% of libraries per condition. Library sizes were scale-normalized by the TMM method using EdgeR software[57] and further subjected to linear model analysis using the voom function in the limma package[58]. Differential expression analysis was performed using DeSeq2[59]. Genes with fold change greater than two and false discovery rate corrected *p*-value (Benjamini-Hochberg procedure) <0.05 were considered to be differentially expressed.

**Identification of daily rhythmic transcripts**. We used two programs, JTK-Cycle[60] and ARSER[61] implemented in MetaCycle[62], an integrated R package with parameters set to fit time-series data to exactly 24-h periodic waveforms. While JTK Cycle uses a non-parametric test called the Jonckheere-Terpstra test to detect rhythmic transcripts[60], ARSER uses "autoregressive spectral estimation to predict an expression profile's periodicity from the frequency spectrum and then models the rhythmic patterns by using a harmonic regression model to fit the time-series"[61]. For both the programs voom-TMM normalized count data was used as input data. A gene was considered cyclic if both the programs identified it as a rhythmic transcript with significance bounded by *p* < 0.05 for the parasite and host rhythms mismatching experiment, where 11 time points separated by 3 h were used and by *q* < 0.05 for the SR10 experiment where 14 time points separated by 3 h were used. We used the data from only 14 out of 17 time points. The first three time-points having been excluded owing to low numbers of mapped reads to *P. chabaudi* (<1 million). For the *P. falciparum* time-series experiment, a gene was considered cyclic if both programs identified it as a rhythmic transcript with significance bounded by *q* < 0.05 where a total of 25 time-points separated by two hours were used. The output from ARSER concerning amplitude, phase and period of rhythmic transcripts was used for further analysis. Median amplitude of rhythmically expressed genes was calculated by taking the amplitude information from ARSER output. Two biological replicates per time point were used for RNAseq analysis for both host rhythm mismatch and *sr10*KO data.

Time points and biological replicates (*n* = 2) were clustered using hierarchical clustering in R software environment with Pearson correlations from normalized count values as input and agglomerative hierarchical clustering function "hclust" (option ward.D2) was used for cluster generation. The linear histogram plots representing phase distribution of cycling transcripts were generated using Oriana (www.kovcomp.co.uk/oriana/). PCA was performed on voom-TMM normalized data using *princomp* in R software environment.

To determine the overall FDR of transcripts with daily rhythms in the host-circadian rhythm mismatching experiment, the time points of collection were randomly permuted 1000 times and the number of transcripts with daily rhythms was assessed for each of the permutations by both the programs used. We observed that in every permutation, the number of rhythmic transcripts detected was less than the observed number of rhythmic transcripts detected when the samples were in correct order of sampling. A similar approach to determine the FDR was used by Rijo-Ferreira et al.[3]. This was done for both matched and mismatched parasite datasets, where *p* < 0.05 was used as a cut-off to identify genes with daily rhythms.

**Analysis of parasite developmental rhythmicity**. The early trophozoite stage was used as the marker stage because it is both easily identifiable and has developmental duration similar to other stages. Rhythmicities in the proportion of parasites at early trophozoite stages was determined by Fourier transformed harmonic regression in Circwave[63]. A cosine wave was fitted to data from each individual infection and compared to a straight line at the mean via an *F*-test. The period was allowed to change between 18 and 30 h (i.e., 24 h ± two sampling periods for the host rhythm mismatch experiment and one sampling period for the analysis of *sr10*KO strains) and the fit was considered significant if the adjusted (to account for multiple tests for different periods) *p* value was greater than the alpha of 0.017. General linear models were used to determine if the characteristics of rhythms varied according to treatment and parasite strain (using the package lme4 in R version 3.4.0).

**Plasmid construction to modify the *sr10* gene locus**. Plasmids were constructed using the MultiSite Gateway cloning system (Invitrogen). For *P. chabaudi*, One thousand base pair long regions at the 5′ and 3′ UTRs of *Pchsr10* were PCR-amplified from *P. chabaudi* with *att*B-flanked primers, *Pchsr10*-5U.B1.F and

*Pch*-5U.B2.R to yield *att*B1-*Pchsr10*-5U-*att*B2 fragment, and *Pchsr10*-3U.B4.F and *Pchsr10*-3U.B1r.R to yield *att*B4-*Pchsr10*-3U-*att*B1r fragment. The *att*B1-*Pchsr10*-5U-*att*B2 and *att*B4-*Pchsr10*-3U-*att*B1r products were then subjected to independent BP recombination reactions with pDONR221 (Invitrogen) and pDONRP4P1R (Invitrogen) to generate pENT12-*Pchsr10*-5U and pENT41-*Pchsr10*-3U entry clones, respectively. All BP reactions were performed using the BP Clonase II enzyme mix (Invitrogen) according to the manufacturer's instructions. pENT12-*Pchsr10*-5U, pENT41-*Pchsr10*-3U, and linker pENT23-3Ty entry plasmids were subjected to LR recombination reaction (Invitrogen) with a destination vector pDST43-HDEF-F3 (that contains the pyrimethamine resistant gene selection cassette *hDHFR*) to yield knockout construct pKO-*Pchsr10*. LR reactions were performed using the LR Clonase II Plus enzyme mix (Invitrogen) according to the manufacturer's instructions. *P. yoelii* 17 × 1.1 pp *sr10* knockout parasites were also generated using the same procedures adopted for *P. chabaudi* using *Pysr10*-5U.B1.F and *Pysr10*-5U.B2.R to yield *att*B1- *Pysr10*-5U-*att*B2 fragment, and *Pysr10*-3U.B4.F PySR10-3U.B1r.R to yield *att*B4- *Pysr10*-3U-*att*B1r fragment from *P. yoelii* 17 × 1.1 pp gDNA. These fragments were then used in independent BP reactions and subsequent LR reactions as above to generate pKO- *Pysr10* construct. All primers used are listed in Supplementary Data 8.

**Parasite transfection**. Schizonts from *P. yoelii* and *P. chabaudi*-infected mice were enriched by centrifugation over a Histodenz density cushion. Histodenz™ (Sigma-Aldrich, St. Louis, MO) solution was prepared as 27.6 g/100 ml in Tris-buffered solution (5 mM Tris-HCl, 3 mM KCl, and 0.3 mM CaNa2-EDTA, pH 7.5) and then diluted with equal volume of RPMI1640-based incomplete medium containing 25 mM HEPES and 100 mg/l of hypoxanthine[64]. The schizont-enriched parasites were transfected using a Nucleofector™ 2b device (Lonza Japan) using 20 µg of linearized plasmids for each transfection[65]. Stable transfectants were selected by oral administration of pyrimethamine (0.07 mg/ml) and cloned by limiting dilution in mice. Stable integration of plasmids in the parasite genomes were confirmed by PCR and sequencing (Supplementary Fig. 7a, b). Genomic forward and reverse primers (F1 and R1) were designed using the conserved 5′ and 3′ UTR regions of *sr10* in *P. chabaudi* and *P. yoelii*. Primers were also designed corresponding to internal sequence regions of drug resistant cassette (R2 and F1) (Supplementary Fig 7a). Amplification from primer pairs F1 and R2 confirmed the 5′ integration of plasmid and from F2 and R2 confirmed the 3′ integration of plasmid in *P. chabaudi* *sr10*KO parasites (Supplementary Fig. 7a) and *P. yoelii* *sr10*KO parasites (Supplementary Fig. 7b). To confirm the absence of *sr10* in the *P. chabaudi* and *P. yoelii* *sr10*KO parasites primer pairs internal to *sr10* sequences of *P. chabaudi* and *P. yoelii* were designed and PCRs were performed on both wild type and *sr10*KO *P. yoelii* and *P. chabaudi* parasites. Supplementary Fig. 7c shows the absence of *sr10* in the sr10KO parasites while a single band of desired size was detected in wild type *P. chabaudi* and *P. yoelii* parasites. All primers used are listed in Supplementary Data 8 and sanger sequening results are provided in Supplementary Data 9.

**Real-time quantitative reverse transcriptase PCR analysis**. Total RNA was treated with TURBO Dnase according to the manufacturer's instructions (Thermo Fischer Scientific) to eliminate DNA contamination. The absence of DNA in RNA samples was confirmed by inability to detect DNA after 40 cycles of PCR with HSP40, family A (PCHAS_0612600) gene primers in a 7900HT fast real-time PCR system (Applied Biosystems) with the following cycling conditions: 95 °C for 30 s followed by 40 cycles of 95 °C 2 s; 60 °C for 25 s. *P. chabaudi* DNA was used as positive control. For both the mismatching and SR10 experiments, gene expression profiles were obtained for a total of 87 genes from eight time-points. Two biological replicates per experimental condition and two technical replicates per biological replicate were run on a Biomark HD microfluidic quantitative RT-PCR platform (Fluidigm) to measure the expression level of genes. For the SR10 experiment, first strand cDNA synthesis was performed using reverse transcription master mix according to the manufacturer's instructions (Fluidigm) and for the mismatching experiment, first strand cDNA synthesis was performed using a High-Capacity cDNA reverse transcription kit according to the manufacturer's instructions (Thermo Fisher Scientific). Pre-amplification of target cDNA was performed using a multiplexed, target-specific amplification protocol (95 °C for 15 s, 60 °C for 4 min for a total of 14 cycles). The pre-amplification step uses a cocktail of forward and reverse primers of targets (genes of interest) under study to increase the number of copies to a detectable level. Products were diluted 5-fold prior to amplification using SsoFast EvaGreen Supermix with low ROX and target specific primers in 96.96 Dynamic arrays on a Biomark HD microfluidic quantitative RT-PCR platform (Fluidigm). Expression data for each gene were retrieved in the form of Ct values. Normalization of transcript expression level was carried out using *P. chabaudi* HSP40, family A (PCHAS_0612600) gene. We chose *P. chabaudi* HSP40 to normalize transcript expression because it was found to arrhythmic in expression as determined from JTK-Cycle[60] and ARSER[61] output (expressed constantly throughout the IDC) in all the time-series RNAseq data from all the strains used in this study, including host rhythm matched and mismatched *P. chabaudi* parasites (Supplementary Data 1 and Supplementary Data 4). For *P. falciparum* the same steps were followed up to cDNA synthesis. The housekeeping gene Seryl tRNA ligase (PF3D7_0717700) was used to check DNA contamination in RNA samples and for normalization of Ct values. Seryl tRNA ligase was used a control gene because it was also found to be arrhythmic in our *P. falciparum* IDC time-series

RNAseq data as determined from JTK-Cycle[60] and ARSER[61] output. We have also verified the expression of both *P. chabaudi* HSP40, family A and *P. falciparum* Seryl tRNA ligase visually and found it to be arrhythmic. Seryl tRNA ligase has also been widely used as a control gene in previous *P. falciparum* studies[66]. Twelve randomly selected genes were initially selected for validating the time-series RNAseq data out of which nine genes gave specific PCR products observed through dissociation curve analysis. These nine genes (Supplementary Fig. 6e) were further used for validating the time-series RNAseq data. qPCR-based quantification was carried out using Fast SyBr green (Thermo) with the following cycling conditions: 95 °C for 30 s followed by 40 cycles of 95 °C 2 s; 55 °C for 30 s, and 60 °C for 25 s. All primers used are listed in Supplementary Data 8.

**Gene ontology enrichment analysis**. *Plasmodium chabaudi* and *P. falciparum* gene ontology terms were downloaded from the UniProt gene ontology annotation database (https://www.ebi.ac.uk/GOA). Genes with daily rhythms were segregated into 12 groups based on their phase of maximum expression as determined from ARSER output and gene ontology enrichment analysis was performed on each groups using GOstats R package[67]. In the case of *P. falciparum*, GO-enrichment analysis was performed on all the identified circadian transcripts. GO terms were considered only if statistical tests showed FDR corrected $p < 0.05$. Odds ratio was calculated by dividing the occurrence for GO term in the input list to the occurrence for GO term in the reference set (i.e., filtered list of all detected genes).

**Identification of differential alternative splicing events**. We considered two consecutive time-points, day 3 PI, time 05.30 GMT (ZT 22.5) and 08.30 (ZT 1.5) to compare between sr10KO and wild type parasites because they follow the time point when genes associated with spliceosome machinery are expressed maximally (ZT 16.5–19.5). In a separate analysis, we compared two consecutive time points (ZT 22.5 and ZT 1.5) within each strain as controls, with an expectation of less differential alternative splicing events within compared to between wild type and *sr10*KO parasites for the same time point.

Differential alternative splicing events in terms of differential exon usage were detected using the DEXSeq program v 1.20.02[68] with modified scripts as reported in Yeoh et al.[69]. The *p* value significance level was set to 0.05 for the identification of differential exon usage. Comparison was made between *P. chabaudi* wild type and the SR10 knock out strains for two time points i.e., Day 3, ZT 21 and Day 3, ZT 0/24 post-infection. For these two time points, RNAseq read depth was increased by performing additional rounds of sequencing in order to detect AS events more reliably. Two biological replicates per time-point were used.

**Reporting summary**. Further information on research design is available in the Nature Research Reporting Summary linked to this article.

## Data availability

RNAseq data sets are available in Gene Expression Omnibus under accession numbers GSE132647, GSE144976. See also Supplementary Data 1, 3, and 5. The source data underlying Figs. 1d, e, f, 3e, f, g, h, 4d, e, Supplementary Figs. 1d, 2c, 6c, d are provided as Source Data file.

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

## Acknowledgements

The project was supported by a faculty baseline fund(BAS/1/1020-01-01) and a Competitive Research Grant (CRG) award from OSR (OSR-2018-CRG6-3392) from the King Abdullah University of Science and Technology (KAUST) to A.P. R.C. is supported by Japanese Society for the Promotion of Science (JSPS), Japan Grant-in-Aid for Scientific Research Nos. 24255009, 25870525, 16K21233, and 19K07526. SER and AJOD are supported by Wellcome (202769/Z/16/Z; 204511/Z/16/Z), the Royal Society (UF110155; NF140517) and the Human Frontier Science Program (RGP0046/2013). The authors thank the staff of the Bioscience Core Laboratory in KAUST for sequencing RNAseq libraries and all members of the Reece lab at the University of Edinburgh and pathogen genomics lab at KAUST for assistance during the experiments. This work was partly conducted at the Joint Usage/Research Center for Tropical Disease, Institute of Tropical Medicine, Nagasaki University, Japan.

## Author contributions

Conceptualization, A.P. and S.E.R.; Methodology, A.P., S.E.R., A.K.S., A.J.O.D., H.M.A., A.R., R.C., and O.K.; Investigation, A.K.S., A.R., A.J.O.D., R.C., and H.M.A.; Formal analysis, A.K.S., A.R., A.J.O.D., H.M.A., A.K., A.M.A.H., F.B.R., and H.R.A.; Writing—Original draft, A.K.S., S.E.R., and A.P.; Writing—Review and Editing, A.K.S., S.E.R., R.C., and A.P.; Funding acquisition, S.E.R. and A.P.; Resources, S.E.R., R.C., and A.P.; Supervision, A.P. All authors read and approved the final manuscript.

## Competing interests

The authors declare no competing interests.
