## [Peer Review File · Nature Communications]

Reviewers' Comments:

Reviewer #1:

Remarks to the Author:

In this manuscript, Subudhi et al study Plasmodium parasite-intrinsic daily rhythms that impact the intraerythrocytic developmental cycle (IDC). They first study the rhythmic transcriptome of *P. chabaudi* in vivo in mice, comparing parasites with rhythms matched or mismatched with the host. Then they compare these findings to the rhythmic transcriptome of the human malaria parasite *P. falciparum*. Many transcripts are found to show 24h rhythms in both species, including in free-running *P. falciparum*, and there is a strong effect of parasite-host mismatch. In the last part of the manuscript, the authors focus on a transcript found to be rhythmic in both models, encoding the GPCR serpentine receptor 10 (SR10). Knocking out SR10 reduced the IDC by 2-3 h. This is an interesting and well-performed study. The discovery of endogenous gene expression rhythms in Plasmodium species is important, and makes it only the second unicellular parasite where such endogenous rhythms have been found (after *Trypanosoma brucei*). The use of 3 Plasmodium species, including mouse and human parasites, and parasites with 24h and 48h IDCs, is a strength of the paper, as is the detailed analysis of the possible role of SR10 in IDC control and host-parasite interaction.

1) One potential issue with the data is the criteria to define the transcripts as rhythmic or not, and the need to ensure that the proportion of rhythmic transcript is not overestimated. They do a resampling analysis to assess the FDR, but the outcome of this analysis is not clear. More details will be needed to judge about the validity of the approach. They could also do the analysis for rhythmicity over a range of different FDR cut-offs, and for example for the matched/mismatched mouse experiment, show that there are always more rhythmic transcripts in the matched parasites. Also, for the different experiments, they should show side by side the heat maps of the exact transcripts in both conditions, so that one can see de visu the loss of rhythmicity, or, on the contrary, any residual rhythmicity.

2) Another potential issue is the classification of transcripts as rhythmic and non-rhythmic, and the conclusions, throughout the manuscript, about the sets of transcript that lose or gain rhythmicity. This is a somewhat arbitrary delineation, that depends on the threshold set for defining rhythmicity. Actually, the guidelines paper of Hughes et al, JBR, 2017, recommended against relying primarily on groups of rhythmic vs arrhythmic data series. Actually, as noted by Subudhi et al, there is a clear effect of amplitude (e.g. Fig 1d), such that much of the loss or gain of rhythmicity might be due to assigning lower amplitude rhythms to an "arrhythmic" category. This is supported by the clear remaining rhythmicity, for example heat maps in Fig. 2b, 4a.

3) Some parts of the manuscript are very hard to understand, e.g. lines 145-150, 177-181, 218-219, 367-370. In many cases, these are important aspects of the authors' interpretation on what might be involved in the clock or what could be under the influence of the host's rhythms, and these parts should be clarified.

4) The mouse/*P. chabaudi* experiments were performed under a light-dark cycle. Therefore, none of the rhythms found in these experiments can be termed "circadian". Use alternate words such as "daily" or "time-dependent".

5) Biological repeats are unclear throughout the manuscript and should be clarified. For example, in the mouse studies, 4 mice per time point were used, but it is unclear how many were used in the RNA-seq experiment, and what replicates are used for the clustering analysis. Please clarify the Results and Methods sections.

6) I understand that the period obtained by the ARSER analysis was used in the discussion (e.g. lines

198-201). But can one really use and trust a period assessed from a data series from only 30 h sampling (i.e. just over one 24h cycle)?

7) I don't feel that the title of the manuscript represents well the content of the study. The study is not only about the effect of "disrupting the host-parasite coordination". The title seems to focus on the effect of this on the IDC duration, which is in my view at best a small portion of the study. On the other hand, what I consider the most important aspects of the findings are not addressed at all in the title, e.g. the endogenous circadian rhythms of gene expression (in particular in *P. falciparum*), and the involvement of SR10.

8) Lines 50-51: References 6-7 did not show "clocks that operate via non-transcriptional oscillators or post-translational control". Tone down the statement.

9) Methods section "Transcriptome sequencing and analysis" (line 796) should be grouped with the "Time series gene expression..." section (line 744). More generally, please ensure that the Methods are organized logically.

10) Lines 822-824: Why use $p < 0.05$ in one experiment and $q < 0.05$ in another?

Reviewer #2:

Remarks to the Author:

Subudhi et al sought to understand the relationship between the Plasmodium intra-erythrocytic developmental cycle (IDC) and the host circadian cycle. They first evaluated time series gene expression data of parasites from infections that were in synchrony and out of synchrony with the host circadian rhythms. They observed several biological processes that seem to be altered by the misalignment with the host cycle. The authors found the same biological processes in *P. falciparum* in *in vitro* conditions, suggesting the role of these pathways specifically in IDC. Using both results they propose SR10 as a receptor that mediates the interaction between host circadian cycle and IDC processes in the parasites. The authors later validate their hypothesis showing the changes in transcriptome and IDC cycles in parasites lacking SR10, showing its importance for the link between host circadian cycle and parasites' endogenous time keeping mechanism of IDC control.

The manuscript is well-written and detailed. The results are well supported by the data and support the authors' conclusions. I would suggest a few minor revisions to clarify some specific points and help readers have a better comprehension.

(Lines 124-126) A figure showing the significance versus the amplitude of cyclic genes would help the reader have a general sense of the effect size of the significant genes.

(Lines 127-132) It is not clear if the $FDR < 0.05$ reported in line 32 reflects the threshold used in each iteration of the permutation process, or if this is the overall false discovery rate discovered when considering all 1000 permutations. A measure like z-score could help the reader identify how far the observed value is from the values obtained in the permutations.

(Pages 7 and 8) The authors claim that it was expected that IDC genes should peak 6 hours later in mismatched parasites (lines 143-149). That would make me think that the set of 685 genes that were later identified as with delayed phase transcriptions (lines 168-169) are the best candidates for possible regulators of IDC in the parasites. From that perspective, I believe that it would be helpful to

provide comparisons of this set of genes with the results from *P. falciparum* and with the results from the SR10 knockouts.

The figure 1d suggests that most rhythmically transcribed genes have smaller amplitudes (~ 1) and a smaller proportion have higher amplitude. The figure suggests that the misalignment affect parasites' transcriptomes by lowering the amplitudes of the cycling genes. It would be interesting to see where the genes with transcription phase delayed of 6 hours are located in this distribution. Since these genes seem to be the ones mostly directed linked with IDC, possibly acting as master regulators or markers of the process, it would be interesting to see if there is a change in their amplitude in the mismatched parasites.

(Lines 185-192) I find very interesting that some genes gain rhythmicity of transcription in mismatched parasites. How the change in light affect molecular functions in the host? A discussion about what might be happening in the host that could trigger transcription related functions in the parasites could provide insights about the mechanistic and evolutionary aspects of host-parasite interactions.

(Lines 373-380) I believe a better description of why the authors picked SR10 for further validation would help in the overall understanding. When describing *P. chabaudi* results, the authors shift the discussion from genes that lose rhythmicity, genes that are cyclic in both matched and mismatched, and genes that have delayed transcription phase. The SR10 was the top ranking gene in the *P. falciparum* results, but how does it fit with these other classes classified in *P. chabaudi*? Was it among the top ranking genes of in one of those sets too? Was it among the genes with delayed transcription phase?

(Lines 488-491) I believe it would be interesting to provide comparisons of the SLCGs genes and the ones with 6 hours delay in the phase of transcription.

(Lines 501-502) An overrepresentation test could provide statistical significance for the observed numbers of spliceosomal proteins in the SLCG set.

(Lines 544-547) Please inform the significance of spearman-rank correlation used in the comparison of real time qPCR and RNA-seq test.

Reviewer #3:

Remarks to the Author:

The parallel timing of the intraerythrocytic (life) cycle of the malaria parasite and the circadian rhythm of its host provides an opportunity to investigate the genetic basis for coordination between the two. Research has already suggested for example that the nutrient environment that the mammalian host provides for the parasite, and its fluctuation over time, can influence the timing of the parasite development cycle. Alternatively, the parasite maintains the timing of its growth cycle independent of circadian host signals.

This manuscript describes a transcriptomics approach to identify parasite genes that are essential to maintaining the synchrony with the host.

Design: Parasites obtained from donor mice one "time-zone" are used to infect mice in the same time-zone ("matched") , as well as another cohort of mice living in a +12hr time zone ("mismatched"). The parasites are given 4 days to recover from jet lag before being sampled every three hours over the

next 30 hrs for RNA-seq, 4 mice per time point.

Major issues:

It would be appropriate to use an FDR adjusted p-value cutoff instead of the raw p-value < 0.05 in the test for 24-hr periodicity, given the great many genes tested. Apparently ARSER method for time-course modeling predicts not just the likelihood of cycling, but the period, amplitude, and phase, as summarized in Fig 1. I think the significance estimate is the goodness of fit, but the period estimate indicates whether the gene fits the 24 hr periodicity, as opposed to a shorter or longer cycle.

Using data in table

2732 (ARSER FDR p-value $< .05$) out of 3784 genes measured in "matched" time course dataset = 72%

Those with an ARSER predicted period between 23.5 and 24.5 hrs. & FDR $< .05$ is 816 (21.5%)

I should think that the definition of a negative result in the test for circadian cycling (in the mismatch group) would include genes that are NOT having an estimated period in a range, eg. 23 – 25 hr, regardless of p-value.

Applying all three criteria

("matched" ARSER FDR p-value $< .05$)

& ("matched" ARSER period between 23.5 and 24.5)

& ("mismatched" ARSER period greater than 25 or less than 23)

Results in 350 genes that did not recover circadian expression pattern 4 days after transfer to a time-shifted host.

Alternatively, a definition of asynchronous expression such as 'significant variation in expression across all the time points, and not significantly circadian' would offer an advantage of having a performance criteria for the denominator that does not include low expression or invariant genes that could be called "failed" tests for circadian rhythm, as opposed to "negative" result.

Figure 1b: The heatmaps should be organized as they are in Figure 2b, to enable direct comparison of the cohorts.

Figure 1d,e,f: similarly, the distributions time series analysis statistics are compared, but it would be helpful to plot the data for individual genes as a scatterplot of matched vs mismatched. For example, genes with high time series amplitude

A couple of examples are attached. It is interesting to see the clusters of genes in the discordant quadrants: most genes that were high period in the matched set were found to be low period in the mismatched, and a bias in the genes that were low period in the matched set to appear as higher period in the mismatched. A longer time course that included 2 and 3 days post subinoculation would confirm a trend and possible role for these out of sync genes.

Figure 2, the organization of circadian genes by phase seems very useful, but only one of the gene set enrichment assignments, Transmembrane transport, is both convincing ($p=4.8E-09$) and meaningful as a life cycle relevant function. Unless these can be

Minor issues:

Figure 1d: y-axis label should be simply "density", or "frequency density".

Figure 1e: lines mentioned in legend are barely visible in the graph

Figure 1f: y-axis label should be simply "density", or "frequency density".

Figure 1f: HCRS is not defined.

GO term enrichment statistics should be calculated using the filtered list of genes described in the RNA-seq methods section (806-807), the detected genes, not the whole genome, as the reference set. The authors should consider setting an upper bound on the on the frequency of the term in the reference set so that the most general GO terms are excluded.

Supp. Pf time course results should include the JTK statistics for all genes.

Many diagrams in the suppl. pdf file have illegible labels. Eg. Dendrograms sup.fig.2, Venn diagrams.

Reference 1 is not available online, would an older reference suffice? Eg. Coordination of circadian timing in mammals. Nature 418, 935–941 (2002).

831: "hierarchical clustering procedure" -> "hierarchical clustering function 'hclust'"

832 : There don't seem to be any circular histogram plots.

Conclusion:

Very large dataset carefully measured and curated. The tools used for data analysis are appropriate but applied such that the statistical analysis is sub-optimal. Despite the large amount of data collected, the complexity of the biological systems mandates that a longer time course be collected to support the authors' claims regarding gene function and biological mechanisms.

Bivariate Fit of period of mismatched By period of matched

.1 .2 .3 .4 .5 .6 .7 .8 .9 Quantile Density Contours

Quantile Density Contours

Variable	Kernel Std
period of matched	0.310438
period of mismatched	0.388543

period of mismatched vs. period of matched

Where($\text{fdr_BH of matched} \geq 0.004833957$ & $\text{fdr_BH of matched} \leq 0.05$)

We thank the reviewers for their time and effort in reviewing our paper now titled “**Malaria parasites regulate the duration of the intraerythrocytic cycle via serpentine receptor 10 and coordinate development with host daily rhythms**”. We have incorporated most of their suggestions and feel that these modifications have improved the paper’s clarity and focus considerably. Please find below our comments and a list of changes to the manuscript. In addition to making changes requested by the reviewers we have also undertaken new analyses of a superior time-series transcriptome dataset, as described below.

General comments and changes to the manuscript

1. To match the prescribed word length of *Nature Communications* for a research manuscript, we have cut down the manuscript from ~6,500 words to ~4,900 words. In order to achieve this, we have edited the introduction, results and discussion section to remove extra information without compromising the clarity of the paper. Particularly, we have reduced the size of the introduction by keeping only the most relevant information and moved additional supporting data either to the supplementary information section or methodology section to reduce the length of the results section. Due to the extensive rewrite, we have focused on highlighting (in blue) alterations to the text requested by the reviewers.
2. We have replaced the *P. falciparum* transcriptome analysis that utilized microarray data to identify daily rhythmic transcripts (from Bozdech et al., 2003, *PLoS Biology*) with our new two-hour resolution time-series RNAseq data from *P. falciparum* intra-erythrocytic developmental stages. By using a new data set, we have been able to incorporate all the revisions to the *P. falciparum* genome annotation that have occurred over the last 17 years. This includes the addition of ~400 new independent genes and modifications to boundaries/structures of about 25% of genes (Gardner et al., 2002 *Nature*). Furthermore, time-series expression data of 894 genes were missing in the previously analysed datasets. For these reasons, we have generated our own biologically reproducible time series expression data with a 2 h resolution. We use these data to identify the free-running rhythmic genes using RNAseq of *P. falciparum* cultured at constant temperature and in dark conditions. Importantly, this has also allowed us to analyse all the updated annotated transcripts (year 2020 annotation) in the *P. falciparum* genome (5,711) compared to only 4,817 genes that were represented by probes in the previously published microarray dataset. As before, we have considered a gene to be rhythmic if both the ARSER and JTK cycle programs predicted it to be so (BH.Q < 0.05). With this stringent criterion, we have identified 361 genes to be rhythmic in *P. falciparum* in free-running conditions (**Figure 3a**). Importantly, consistent with our original analysis, *sr10* was identified as one of the top-ranked 24h rhythmic genes (rank 28 based on JTK cycle q value) in the new time-series RNAseq dataset.

To validate the accuracy of *P. falciparum* time-series RNAseq datasets, the expression profiles of nine randomly picked genes were validated using real-time qPCR. Expression profiles of genes from real-time qPCR and RNAseq data were found to be tightly correlated (**Supplementary Fig. 6e**).

We found biological replicates to be tightly clustered (**Figure R1a**). We also found a good overall correlation between the expression patterns of genes from microarray and RNAseq data (average Pearson correlation of 4817 genes = 0.603) (**Figure R1b and R1c for reviewers**). When we compared the 361 free-running rhythmic genes identified from our new time-series RNAseq data with the 494 genes that were identified from the previously published time-series microarray data, 93 genes were found to be common between them (**Figure R1d for reviewers**). These comparisons are illustrated below (Fig R1) for the reviewers only. In the revised manuscript, we have included results only from the time-series RNAseq dataset, as we believe that RNAseq time-series from this study is totally independent and the expression patterns of nine randomly picked genes were verified using real-time qPCR showing very high correlation between the independent datasets (**Supplementary Fig. 6e**).

Figure R1 for reviewers. a) Clustered dendrogram of 25 time points with two biological replicates per time point using hierarchical clustering algorithm. Clustering was done using normalized count data. b) Heat maps of genes (n=2795) in the free-running IDC cycle from Bozdech *et al.*, 2003 microarray dataset (heat map on left panel) and RNASeq from this study (heat map on right panel) at 2 h resolution. Gene expression values have been centered and normalized as Z scores. Genes were ordered based on *P. falciparum* phaseogram shown in Llinas *et al.*, 2006, *Nucleic Acids Research* c) Pearson correlation values of expression of 4,817 between

microarray and RNASeq data sets. d) Venn diagram showing comparison of identified daily rhythmic genes from RNASeq data set and microarray data set.

3. We have also included a new PCR genotyping figure (Supplementary Fig 7) and Sanger sequencing results (Supplementary Data 9) confirming the successful knockout of *sr10* in two *P. chabaudi sr10KO* clones (A and B used in phenotype experiments) and the *P. yoelii sr10KO* clone. This is in addition to the Supplementary Fig. 3c where RNAseq data confirms the deletion of *sr10* in *P. chabaudi sr10KOA* clone used for daily rhythmic transcriptome analysis.
4. In addition, we have overhauled the elements contained in the figures, based on comments from reviewers and incorporated our new analyses, summarized as:
 - Fig. 1b (modified heatmaps)
 - Fig. 1d (modified density plot)
 - Fig. 1e (Thicker lines representing mean and standard deviation)
 - Fig. 3a (new *P. falciparum* time-series heatmap)
 - Fig. 3b (new gene ontology enrichment analysis of *P. falciparum* 24h rhythmic data)
 - Fig. 4a (modified heatmaps)
 - Supplementary Fig. 1e (new line graph)
 - Supplementary Fig. 2d (new scatter plot representing amplitudes of genes)
 - Supplementary Fig. 6e (dot plot showing the correlation of expression of 9 genes measured using RNASeq and qPCR from *P. falciparum* time-series experiment.
 - Supplementary Fig. 7 (new figures confirming successful knockout of *sr10* in *P. chabaudi* and *P. yoelii sr10KO* parasites.

We believe that the new analyses, figures, and improvements suggested by the reviewers have resulted in an accurate and complete picture of the IDC rhythms of malaria parasites.

Reviewer #1 (Remarks to the Author):

In this manuscript, Subudhi et al study Plasmodium parasite-intrinsic daily rhythms that impact the intraerythrocytic developmental cycle (IDC). They first study the rhythmic transcriptome of *P. chabaudi* in vivo in mice, comparing parasites with rhythms matched or mismatched with the host. Then they compare these findings to the rhythmic transcriptome of the human malaria parasite *P. falciparum*. Many transcripts are found to show 24h rhythms in both species, including in free-running *P. falciparum*, and there is a strong effect of parasite-host mismatch. In the last part of the manuscript, the authors focus on a transcript found to be rhythmic in both models, encoding the GPCR serpentine receptor 10 (SR10). Knocking out SR10 reduced the IDC by 2-3 h. This is an interesting and well-performed study. The discovery of endogenous gene expression rhythms in Plasmodium species is important, and makes it only the second unicellular parasite where such endogenous rhythms have been found (after *Trypanosoma brucei*). The use of 3 Plasmodium species, including mouse and human parasites, and parasites with 24h and 48h IDCs, is a strength of the paper, as is the detailed analysis of the possible role of SR10 in IDC control and host-parasite interaction.

1) One potential issue with the data is the criteria to define the transcripts as rhythmic or not, and the need to ensure that the proportion of rhythmic transcript is not overestimated. They do a resampling analysis to assess the FDR, but the outcome of this analysis is not clear. More details will be needed to judge about the validity of the approach. They could also do the analysis for rhythmicity over a range of different FDR cut-offs, and for example for the matched/mismatched mouse experiment, show that there are always more rhythmic transcripts in the matched parasites. Also, for the different experiments, they should show side by side the heat maps of the exact transcripts in both conditions, so that one can see de visu the loss of rhythmicity, or, on the contrary, any residual rhythmicity.

Response: Ideally, we would like to have used the FDR adjusted p-values provided by the algorithms to detect 24h rhythmic genes. However, these algorithms are known to be conservative, especially at lower sampling rates. Because of the high cost of sequencing, our study was performed using a three hour sampling rate, and thus these algorithms produce adjusted p-values that are generally too high (see Wu et al. 2014 <http://doi.org/10.1177/0748730414537788>; Hutchison et al. 2015 <http://doi.org/10.1371/journal.pcbi.1004094> and Rijo-Ferreira et al., 2017 <http://dx.doi.org/10.1038/nmicrobiol.2017.32>).

Due to this, we were unable to use BH or Bonferroni adjusted p-values, and instead used a similar approach to that of Koike et al. 2012 (<http://doi.org/10.1126/science.1226339>) and Rijo-Ferreira et al., 2017 (<http://dx.doi.org/10.1038/nmicrobiol.2017.32>) where the unadjusted p-values from multiple algorithms are used, each based on distinct statistical methods, and selection is performed based on the agreement between methods. In order to balance the cost and reproducibility, we chose to use two biological replicates per time-point keeping sampling rate at three hours. Common rhythmic genes from two different algorithms from two biological replicates further increased the confidence in detecting rhythmic transcripts, and we are confident that we have not overestimated the numbers.

To further evaluate the false-discovery rate, we estimated an empirical FDR using a permutations test. As can be seen in the Supplementary Fig. 1b, when the sampling order was permuted 1,000 times, it always gave a smaller number of rhythmic genes compared to observed rhythmic genes predicted when the samples were kept in correct order of collection with an overall FDR < 0.05 for all permutations tests conducted for each algorithm used for each experimental condition, a similar approach was also used by Rijo-Ferreira et al., 2017 (<http://dx.doi.org/10.1038/nmicrobiol.2017.32>).

We believe that all these analyses provide enough confidence that the proportion of rhythmic transcripts reported is not overestimated. We have also calculated the Z score to identify how far the observed value is from the values obtained in the permutations suggested by reviewer 2. The Z scores were found to be too high and were between 16.68, 6.86, 13 and 5.50 for all 4 permutations processes performed for matched JTK, matched ARSER, mismatched JTK and mismatched ARSER respectively. The Z scores show that our observations are far higher than the distribution of permutations.

We also find high similarity in the daily rhythmic genes detected ($p < 0.05$, 11 time points, 3 h resolution data 2 replicates) in the host rhythm matched parasites with the daily rhythmic genes detected in the wild type parasites in the SR10KO experiment (BH.Q < 0.05; 14 time points, 3 h resolution data, 2 replicates) (Supplementary Data1 and Supplementary Data5). Note, these groups of parasites are analogous because they constitute “control infections” in which hosts experienced 12:12 L:D schedules, were fed nocturnally, and in which the parasites were matched to host rhythms. At the phenotypic level, this type of infection is highly repeatable. We found that ~ 84 % of the genes detected as rhythmic in host-rhythm matched parasites ($p < 0.05$) were also detected as rhythmic (BH.Q < 0.05) in wild type *P. chabaudi* parasites. This suggests that rhythmic genes detected in both groups can be considered rhythmic even if they have raw p-value < 0.05 (cutoff considered for analysis in host rhythms mismatch experiment).

As suggested by Reviewer #1, we have now included the heatmaps of the exact transcripts in host-rhythm matched and mismatched parasites to gauge the level of loss of rhythmicity (Figure 1b). We have also included Supplementary Fig. 1e showing the number of daily rhythmic transcripts detected in both matched and mismatched parasites over a range of different FDR cut-offs, as suggested (Line 112-114). It is clear from the figure that there were always more daily rhythmic transcripts in matched parasites compared to mismatched parasites (in other words, there is a loss of rhythmic expression of many genes in mismatched parasites which were rhythmic in matched parasites).

2) Another potential issue is the classification of transcripts as rhythmic and non-rhythmic, and the conclusions,

throughout the manuscript, about the sets of transcript that lose or gain rhythmicity. This is a somewhat arbitrary delineation, that depends on the threshold set for defining rhythmicity. Actually, the guidelines paper of Hughes et al, JBR, 2017, recommended against relying primarily on groups of rhythmic vs arrhythmic data series. Actually, as noted by Subudhi et al, there is a clear effect of amplitude (e.g. Fig 1d), such that much of the loss or gain of rhythmicity might be due to assigning lower amplitude rhythms to an "arrhythmic" category. This is supported by the clear remaining rhythmicity, for example heat maps in Fig. 2b, 4a.

Response: We agree with the reviewer that based on a hard threshold it is sometimes difficult to classify genes as rhythmic or non-rhythmic (some genes might still have residual rhythmicity). In our case, as suggested, we have accounted for the amplitude of genes that remained rhythmic in four matched and mismatched and wild-type and *sr10KO* parasites. We have revised figure 1d to include the amplitude of daily rhythmic genes from both host rhythm matched and mismatched parasites with 6 h delayed phase of expression in the mismatched parasites. Overall, we saw a loss of rhythmicity based on the hard threshold and also a decrease in amplitude of rhythmic genes in host rhythm mismatched parasites compared to host rhythm matched parasites implicating the role of host rhythms on parasite developmental rhythms. Our aim is to show that when the parasite's rhythm is mismatched to the host's rhythm many genes lose rhythmicity or their amplitude of expression gets dampened severely, rather than quantify the details of changes in expression patterns.

In the revised manuscript, we have also included a new figure (**Supplementary Fig. 2d**) showing a scatter plot of the amplitude of rhythmic genes in matched vs mismatched parasites. Similar to Figure 1d, the majority of genes with high amplitude in host-rhythm matched parasites were found to have lower amplitude in host rhythm mismatched parasites.

3) Some parts of the manuscript are very hard to understand, e.g. lines 145-150, 177-181, 218-219, 367-370. In many cases, these are important aspects of the authors' interpretation on what might be involved in the clock or what could be under the influence of the host's rhythms, and these parts should be clarified.

Response: Apologies for the confusion. These sections have either been modified or removed, and the discussion of what our results mean in the context of a parasite clock is now a focus of the revised discussion section.

4) The mouse/*P. chabaudi* experiments were performed under a light-dark cycle. Therefore, none of the rhythms found in these experiments can be termed "circadian". Use alternate words such as "daily" or "time-dependent".

Response: We agree with the reviewer's point. Parasites may be coordinated to rhythms run by the host's circadian clock or to host rhythms that simply respond to light-dark schedules. We have used the terms 24h or daily where appropriate, instead of circadian.

5) Biological repeats are unclear throughout the manuscript and should be clarified. For example, in the mouse studies, 4 mice per time point were used, but it is unclear how many were used in the RNA-seq experiment, and what replicates are used for the clustering analysis. Please clarify the Results and Methods sections.

Response: For phenotypic results, four replicates were used for each group of host rhythm matched- and mismatched infections, and three replicates per group were used for *sr10KO* experiments. For RNAseq analysis (Lines 262-263, Lines 642-643), we used two biological replicates for all groups in all experiments. We have clarified sample sizes throughout the results and methods sections.

6) I understand that the period obtained by the ARSER analysis was used in the discussion (e.g. lines 198-201). But can one really use and trust a period assessed from a data series from only 30 h sampling (i.e. just over one 24h cycle)?

Response: *P. chabaudi* has a ~ 24-hour intra-erythrocytic developmental cycle with tight synchronicity between parasites during development. Thus, our 30h window captured a complete IDC. Sampling for longer would have compromised our ability to meaningfully compare matched and mismatched parasites because the mismatched parasites would have become further rescheduled to match host rhythms. However, we agree with the reviewer that period measures from a single IDC cycle are estimates rather than precise measurements and we now emphasize this point.

7) I don't feel that the title of the manuscript represents well the content of the study. The study is not only about the effect of "disrupting the host-parasite coordination". The title seems to focus on the effect of this on the IDC duration, which is in my view at best a small portion of the study. On the other hand, what I consider the most important aspects of the findings are not addressed at all in the title, e.g. the endogenous circadian rhythms of gene expression (in particular in *P. falciparum*), and the involvement of SR10.

Response: We have changed to the title to read: “**Malaria parasites regulate the duration of the intraerythrocytic cycle via serpentine receptor 10 and coordinate development with host daily rhythms**”

8) Lines 50-51: References 6-7 did not show "clocks that operate via non-transcriptional oscillators or post-translational control". Tone down the statement.

Response: We have made the introduction more concise and focused, and the discussion of various types of timekeeping has been taken out in the revised version of the manuscript.

9) Methods section "Transcriptome sequencing and analysis" (line 796) should be grouped with the "Time series gene expression..." section (line 744). More generally, please ensure that the Methods are organized logically.

Response: In the revised version, different sub-sections in the method section have been re-arranged to make it more coherent.

10) Lines 822-824: Why use $p < 0.05$ in one experiment and $q < 0.05$ in another?

Response: As mentioned in response to comment 1, when we used the $q < 0.05$ as a cut-off in the host-rhythm matched and mismatched experiment, we observed less than the expected number of genes passing the cut-off in host-rhythm matched parasites compared to wild type parasites in the *sr10KO* experiment (these are analogous treatment groups) where $q < 0.05$ was used as cut-off. Detection of fewer rhythmic genes with $q < 0.05$ in host rhythm mismatched parasites could be due to various reasons: 1) different lab environments 3) sampling time differences 4) sample collection procedures. Most importantly, when we compared the daily rhythmic genes identified in host-rhythm matched parasites with $p < 0.05$ and wild type parasites in the SR10KO experiment with $q < 0.05$, 84% of the daily rhythmic genes ($p < 0.05$) in host rhythm matched parasites were also identified as daily rhythmic in wild parasites ($q < 0.05$) suggesting that for the host rhythm matched and mismatched experiment a cut-off of $p < 0.05$ is reliable.

Reviewer #2 (Remarks to the Author):

Subudhi et al sought to understand the relationship between the Plasmodium intra-erythrocytic developmental cycle (IDC) and the host circadian cycle. They first evaluated time series gene expression data of parasites from infections that were in synchrony and out of synchrony with the host circadian rhythms. They observed several biological processes that seem to be altered by the misalignment with the host cycle. The authors found the same biological processes in *P. falciparum* in in vitro conditions, suggesting the role of these pathways specifically in IDC. Using both results they propose SR10 as a receptor that mediates the interaction between host circadian

cycle and IDC processes in the parasites. The authors later validate their hypothesis showing the changes in transcriptome and IDC cycles in parasites lacking SR10, showing its importance for the link between host circadian cycle and parasites' endogenous time keeping mechanism of IDC control.

The manuscript is well-written and detailed. The results are well supported by the data and support the authors' conclusions. I would suggest a few minor revisions to clarify some specific points and help readers have a better comprehension.

1) (Lines 124-126) A figure showing the significance versus the amplitude of cyclic genes would help the reader have a general sense of the effect size of the significant genes.

Response: We are not quite sure what this suggested figure would consist of but we have added a scatter plot of the amplitudes of rhythmic genes in matched vs mismatched parasites to Supplementary Fig. 2, in accordance with comment 2 from reviewer #1. The effects size on amplitude is illustrated in the figure.

2) (Lines 127-132) It is not clear if the $FDR < 0.05$ reported in line 32 reflects the threshold used in each iteration of the permutation process, or if this is the overall false discovery rate discovered when considering all 1000 permutations. A measure like z-score could help the reader identify how far the observed value is from the values obtained in the permutations.

Response: The $FDR < 0.05$ is the overall false discovery rate when considering all 1,000 permutations. We have also calculated the Z score as suggested and included them in Supplementary Fig. 1b and in the text (Lines 108-111). Specifically, the Z scores were between 16.68, 6.86, 13 and 5.50 for all 4 permutations process performed for matched JTK, matched ARSER, mismatched JTK and mismatched ARSER respectively. The Z scores show that our observations are far higher than the distribution of permutations.

3) (Pages 7 and 8) The authors claim that it was expected that IDC genes should peak 6 hours later in mismatched parasites (lines 143-149). That would make me think that the set of 685 genes that were later identified as with delayed phase transcriptions (lines 168-169) are the best candidates for possible regulators of IDC in the parasites. From that perspective, I believe that it would be helpful to provide comparisons of this set of genes with the results from *P. falciparum* and with the results from the SR10 knockouts.

Response: We agree that these genes could be possible regulators of the IDC schedule but the vast majority (if not all) are genes that play a stage specific developmental role and simply reflect parasite development. However, based on this suggestion, we compared the set of 685 genes with delayed phase of expression in mismatched parasites with *P. chabaudi* orthologues of *P. falciparum* rhythmic genes; 36 genes are common between them. Comparing the SLRGs from SR10 knockout experiments with *P. chabaudi* orthologues of *P. falciparum* rhythmic genes; 65 (6%) of the SLRGs are common between them. However, GO analysis of the common genes did not reveal enrichment to any GO term categories. Identification of only a few common genes between different genes sets and the lack of enrichment of any GO terms in the common gene sets suggest that IDC driven and the associated stage specific developmental processes and host circadian rhythm driven parasite processes are different from each other.

4) The figure 1d suggests that most rhythmically transcribed genes have smaller amplitudes (~ 1) and a smaller proportion have higher amplitude. The figure suggests that the misalignment affect parasites' transcriptomes by lowering the amplitudes of the cycling genes. It would be interesting to see where the genes with transcription phase delayed of 6 hours are located in this distribution. Since these genes seem to be the ones mostly directed linked with IDC, possibly acting as master regulators or markers of the process, it would be interesting to see if there is a change in their amplitude in the mismatched parasites.

Response: Almost all cycling genes (including genes with a transcription phase delayed by six hours in mismatched parasites compared to matched parasites) showed lower amplitude in mismatched compared to matched parasites (Lines 153-154). Median amplitudes of the cycling genes in matched parasites that show a transcriptional phase delay of six hours in mismatched parasites are 1.49 and 0.95 respectively. We have modified **Fig. 1d** to illustrate this by including the amplitudes of genes with transcription phases delayed by six hours in both matched and mismatched parasites.

5) (Lines 185-192) I find very interesting that some genes gain rhythmicity of transcription in mismatched parasites. How the change in light affect molecular functions in the host? A discussion about what might be happening in the host that could trigger transcription related functions in the parasites could provide insights about the mechanistic and evolutionary aspects of host-parasite interactions.

Response: Why a few hundred genes should gain rhythmicity in mismatched parasites is intriguing to us too. All hosts were entrained to their light dark schedule before infection, and so we doubt that the hosts mounted different light-induced responses. We propose that the parasite may be compensating for perturbations due to misalignment of the IDC with host rhythms. We also mention the possibility that the parasite possesses an alternate set of rhythmic genes but feel further discussion of these ideas is too speculative.

6) (Lines 373-380) I believe a better description of why the authors picked SR10 for further validation would help in the overall understanding. When describing *P. chabaudi* results, the authors shift the discussion from genes that lose rhythmicity, genes that are cyclic in both matched and mismatched, and genes that have delayed transcription phase. The SR10 was the top ranking gene in the *P. falciparum* results, but how does it fit with these other classes classified in *P. chabaudi*? Was it among the top ranking genes of in one of those sets too? Was it among the genes with delayed transcription phase?

Response: In addition to SR10 having a free running daily rhythm in *P. falciparum*, we have also mentioned that its ortholog in *P. chabaudi* (PCHAS_1433600) showed daily rhythmicity of transcription in both matched and mismatched parasites. SR10 was not a top-ranked rhythmic gene in *P. chabaudi*, which may be due to the high rhythmicity of IDC developmental genes of a 24h parasite overshadowing the rhythmicity of SR10. Finally, SR10 was not amongst the genes with delayed transcription phase in *P. chabaudi*. If SR10 was amongst the genes with delayed transcription phase in *P. chabaudi* this would suggest it is associated with IDC developmental progression rather than the timing of the IDC. This is because a parasite gene that is sensitive to the phase (timing) of host rhythms should follow a pattern correlated with the host's light/dark and feeding schedule. This is what we observe: SR10 expression peaks around ZT14 in both matched and mismatched parasites.

7) (Lines 488-491) I believe it would be interesting to provide comparisons of the SLCGs genes and the ones with 6 hours delay in the phase of transcription.

Response: Please see response to reviewer 2, comment 3

8) (Lines 501-502) An overrepresentation test could provide statistical significance for the observed numbers of spliceosomal proteins in the SLCG set.

Response: We have taken the whole SLCG set as a single gene list to perform GO enrichment analysis using a hypergeometric test. We found that GO term "mRNA splicing via spliceosome" is significantly enriched ($q < 0.005$), as expected (Line 393). We have now included the statistical significance value for this term.

9) (Lines 544-547) Please inform the significance of spearman-rank correlation used in the comparison of real time qPCR and RNA-seq test.

Response: All the Spearman rank correlation values had $p < 0.05$. We have now included this information (Line 433).

Reviewer #3 (Remarks to the Author):

The parallel timing of the intraerythrocytic (life) cycle of the malaria parasite and the circadian rhythm of its host provides an opportunity to investigate the genetic basis for coordination between the two. Research has already suggested for example that the nutrient environment that the mammalian host provides for the parasite, and its fluctuation over time, can influence the timing of the parasite development cycle. Alternatively, the parasite maintains the timing of its growth cycle independent of circadian host signals. This manuscript describes a transcriptomics approach to identify parasite genes that are essential to maintaining the synchrony with the host.

Design: Parasites obtained from donor mice one “time-zone” are used to infect mice in the same time-zone (“matched”), as well as another cohort of mice living in a +12hr time zone (“mismatched”). The parasites are given 4 days to recover from jet lag before being sampled every three hours over the next 30 hrs for RNA-seq, 4 mice per time point.

Response: We wish to clarify that the intention behind choosing 4 days was to allow parasites to replicate to sufficient densities for phenotyping but not have recovered enough to be aligned with host rhythms.

Major issues:

It would be appropriate to use an FDR adjusted p-value cutoff instead of the raw p-value < 0.05 in the test for 24-hr periodicity, given the great many genes tested. Apparently ARSER method for time-course modeling predicts not just the likelihood of cycling, but the period, amplitude, and phase, as summarized in Fig 1. I think the significance estimate is the goodness of fit, but the period estimate indicates whether the gene fits the 24 hr periodicity, as opposed to a shorter or longer cycle.

Using data in table

2732 (ARSER FDR p-value $< .05$) out of 3784 genes measured in “matched” time course dataset = 72%

Those with an ARSER predicted period between 23.5 and 24.5 hrs. & FDR $< .05$ is 816 (21.5%)

I should think that the definition of a negative result in the test for circadian cycling (in the mismatch group) would include genes that are NOT having an estimated period in a range, eg. 23 – 25 hr, regardless of p-value.

Applying all three criteria

(“matched” ARSER FDR p-value $< .05$)

& (“matched” ARSER period between 23.5 and 24.5)

& (“mismatched” ARSER period greater than 25 or less than 23)

Results in 350 genes that did not recover circadian expression pattern 4 days after transfer to a time-shifted host.

Alternatively, a definition of asynchronous expression such as ‘significant variation in expression across all the time points, and not significantly circadian’ would offer an advantage of having a performance criteria for the denominator that does not include low expression or invariant genes that could be called “failed” tests for circadian rhythm, as opposed to “negative” result.

Response: Please see response to Reviewer 1, comment 1, and Reviewer 2 comment 4 where we have clarified why p values were considered instead of FDR adjusted p -values for identifying daily rhythmic genes. Overall we saw a loss of rhythmicity based on a hard threshold and also a decrease in amplitude of rhythmic genes in host rhythm mismatched parasites compared to host rhythm matched parasites, implicating the role of host rhythms on parasite developmental rhythms. Our aim is to show that when parasite rhythm is mismatched with

the host rhythm many genes lose rhythmicity or their amplitude of expression is severely affected rather than quantify these changes in detail.

Figure 1b: The heatmaps should be organized as they are in Figure 2b, to enable direct comparison of the cohorts.

Response: Based on comment 1 from the Reviewer 1, we have kept the expression heat maps of rhythmic genes that were affected in mismatched parasites. Genes are arranged in the same order based on their phase of expression in host-rhythm matched parasites, to enable better visualization of the impact of mismatching the parasite to host daily rhythms.

Figure 1d,e,f: similarly, the distributions time series analysis statistics are compared, but it would be helpful to plot the data for individual genes as a scatterplot of matched vs mismatched. For example, genes with high time series amplitude

Supplementary Fig. 2d of the revised manuscript. Scatter plot between amplitude of daily rhythmic genes from host rhythm matched and mismatched parasites with high time series amplitude (> 1) in host rhythm matched parasites (N=817). Genes in host rhythm mismatched parasites are sorted in descending order based on amplitude.

From the **Supplementary Fig. 2d**, we believe it is clear that the misalignment of the parasite with the host daily rhythms had an overall impact on the parasite's rhythmic transcriptome. The majority of genes with high time series amplitude in host-rhythm matched parasites were found to be with low time series amplitude in host rhythm mismatched parasites.

A couple of examples are attached. It is interesting to see the clusters of genes in the discordant quadrants: most genes that were high period in the matched set were found to be low period in the mismatched, and a bias in the genes that were low period in the matched set to appear as higher period in the mismatched. A longer time course that included 2 and 3 days post subinoculation would confirm a trend and possible role for these out of sync genes.

Response: Please see the response to Reviewer 1, comment 6. We agree that longer time-series data would be beneficial, but we are constrained by the biology of the parasite; finding a balance between allowing infections to run long enough before sampling to ensure enough material to assay but not for so long that the IDC has fully rescheduled and prevents a meaningful comparison of matched versus mismatched parasites.

Figure 2, the organization of circadian genes by phase seems very useful, but only one of the gene set enrichment assignments, Transmembrane transport, is both convincing ($p=4.8E-09$) and meaningful as a life cycle relevant function. Unless these can be

Response: This appears to be an unfinished comment so we await clarification.

Minor issues:

- a) Figure 1d: y-axis label should be simply “density”, or “frequency density”.
- b) Figure 1e: lines mentioned in legend are barely visible in the graph
- c) Figure 1f: y-axis label should be simply “density”, or “frequency density”.
- d) Figure 1f: HCRS is not defined.

Response: All changed /corrected as suggested.

e) GO term enrichment statistics should be calculated using the filtered list of genes described in the RNA-seq methods section (806-807), the detected genes, not the whole genome, as the reference set. The authors should consider setting an upper bound on the on the frequency of the term in the reference set so that the most general GO terms are excluded.

Response: We observed that there were minor or no differences between GO term enrichment statistics when we took total genes in the genome as a reference set compared to a filtered list of genes as the reference set. This could be because the filtered list is ~300 genes less than the genes in the whole genome. We have now updated the GO term enrichment statistics in the supplementary data representing the results of the analysis which were determined by taking only the filtered list of genes as reference set.

In all the cases of GO analysis, we have curated the enriched GO terms manually to represent the short list of terms used in the Figures. We have removed redundant terms wherever possible. However, we have provided all the enriched GO terms in the supplementary data to show the complete pictures of the analyses. While curating the enriched GO terms manually, we excluded general terms for which there was a more specific term available and in the absence of a more specific GO term, we kept the general enriched GO term.

f) Supp. Pf time course results should include the JTK statistics for all genes.

Response: JTK statistics for all genes have been provided in the revised version of the Supplementary Table

g) Many diagrams in the suppl. pdf file have illegible labels. Eg. Dendrograms sup.fig.2, Venn diagrams.

Response: We have modified the diagrams in supplementary figures to make the labels visible.

h) Reference 1 is not available online, would an older reference suffice? Eg. Coordination of circadian timing in mammals. Nature 418, 935–941 (2002).

Response: We have revised the manuscript to meet the word limit criteria provided by *Nature Communications*. During this process we have reduced the total length of the submitted manuscript from ~6,500 words to ~4,900 words in the current revised manuscript. During this process, we have removed reference 1.

831: “hierarchical clustering procedure” -> “hierarchical clustering function ‘hclust’”

832 : There don't seem to be any circular histogram plots.

Response: Typos corrected.

Reviewers' Comments:

Reviewer #1:

Remarks to the Author:

I thank the authors for their careful revision in response to my comments. The manuscript has been improved by these revisions and by the new data and analyses that were included. I only have a few remaining specific comments:

1) One of my main concerns in the previous round of review was about assigning transcripts to rhythmic vs. non-rhythmic groups, and consequently those that remain rhythmic vs. those that lose rhythmicity. The authors provided a detailed response and have added some of the suggested analyses, which I thank them for. However, I would like to see in the manuscript a note of caution about this. It could be for example in the first instance where data are used to discuss a "loss of rhythmicity": page 6, line 167: "Thirty-three percent of genes (n = 1,765) lose rhythmicity of transcription in mismatched parasites." would benefit from being revised to say, instead simply stating a "loss of rhythmicity", that they fell under the threshold for rhythmicity.

2) In many places throughout the manuscript, the authors refer to "rhythmically transcribed genes", "rhythmicity of transcription", "transcription profiles", etc., to refer to the rhythms they find in their analyses. I don't feel that any evidence is provided that shows that the daily rhythms found in the transcripts in this study are due to a rhythmic regulation of transcription (the rhythmic regulation could be all or in part, at the post-transcriptional level). I would suggest revising the manuscript throughout to avoid assuming this. Wording such as "rhythmically expressed genes", "expression profile", etc. could be used instead.

3) It is unclear to me how the authors define and calculate amplitude. For example, on lines 267-268, the authors compare the median amplitudes of *P. falciparum* free-running genes (0.93) with those of *P. chabaudi* rhythmic genes (1.22). What do these values mean, and from what data are they calculated? Explanation in methods would be useful. The depiction of amplitude in Fig. 3c is not helpful because it is a simple arrow that seems to span larger than the rhythm itself (maybe this depiction in the figure should be amended too).

4) Line 293: For clarity, I recommend adding to this sentence a reference to Fig. 3c (e.g., right after "only Pfscr10 showed a 24 h transcription profile").

5) While most of the manuscript talks of "wild type parasites" (for the controls in the sr10 KO experiments), some text in the methods, figure legends and figures only use the term "wild". This should be correct (put "wild type" everywhere).

6) Line 494: As far as I know, all 3 references (46-48) are in mice. To fit with the sentence "These processes are considered central (...) in other taxa.", including additional references backing the statement but in other taxa than mice, would be good.

7) Line 715: "absence" instead of "absent".

8) Lines 747-750: Please explain how HSP40 was checked to be non-rhythmic in *P. Chabaudi* (and thus, suitable to be used as a control gene for rhythmic gene expression analyses), and similarly, provide evidence that Seryl tRNA ligase is non-rhythmic in *P. falciparum*.

Nicolas Cermakian

Reviewer #2:

Remarks to the Author:

I am happy with the rebuttal letter and revised manuscript.

Reviewer #3:

Remarks to the Author:

The revised manuscript is significantly improved and all of my concerns have been adequately addressed.

We thank Nicolas Cermakian and the two other anonymous reviewers for their time and effort in reviewing our manuscript. Their insight and expertise have greatly helped to improve the clarity of the manuscript. Reviewers 2 and 3 do not recommend any further changes and we have provided point-by-point response to the remaining comments of Reviewer #1 (Nicolas Cermakian).

Reviewer #1 (Remarks to the Author):

I thank the authors for their careful revision in response to my comments. The manuscript has been improved by these revisions and by the new data and analyses that were included. I only have a few remaining specific comments:

1) One of my main concerns in the previous round of review was about assigning transcripts to rhythmic vs. non-rhythmic groups, and consequently those that remain rhythmic vs. those that lose rhythmicity. The authors provided a detailed response and have added some of the suggested analyses, which I thank them for. However, I would like to see in the manuscript a note of caution about this. It could be for example in the first instance where data are used to discuss a "loss of rhythmicity": page 6, line 167: "Thirty-three percent of genes (n = 1,765) lose rhythmicity of transcription in mismatched parasites." would benefit from being revised to say, instead simply stating a "loss of rhythmicity", that they fell under the threshold for rhythmicity.

Response: We have revised the sentence as suggested.

2) In many places throughout the manuscript, the authors refer to "rhythmically transcribed genes", "rhythmicity of transcription", "transcription profiles", etc., to refer to the rhythms they find in their analyses. I don't feel that any evidence is provided that shows that the daily rhythms found in the transcripts in this study are due to a rhythmic regulation of transcription (the rhythmic regulation could be all or in part, at the post-transcriptional level). I would suggest revising the manuscript throughout to avoid assuming this. Wording such as "rhythmically expressed genes", "expression profile", etc. could be used instead.

Response: Good point. We have incorporated the suggested changes throughout the manuscript.

3) It is unclear to me how the authors define and calculate amplitude. For example, on lines 267-268, the authors compare the median amplitudes of *P. falciparum* free-running genes (0.93) with those of *P. chabaudi* rhythmic genes (1.22). What do these values mean, and from what data are they calculated? Explanation in methods would be useful. The depiction of amplitude in Fig. 3c is not helpful because it is a simple arrow that seems to span larger than the rhythm itself (maybe this depiction in the figure should be amended too).

Response: Amplitude of each gene was obtained from ARSER output. Further, median of amplitude of all the *P. falciparum* free-running genes and *P. chabaudi* rhythmic genes was calculated by taking the amplitude of each rhythmically

expressed gene from ARSER output. This information has been added in the methods section of the revised manuscript. The arrows depicting the amplitude, phase and period in Fig.3c were provided to help the readers with no or little knowledge in chronobiology terminologies. However, we now understand that this might create confusion and have amended the Fig. 3c to remove the arrows depicting amplitude, phase and period.

4) Line 293: For clarity, I recommend adding to this sentence a reference to Fig. 3c (e.g., right after " only Pfsr10 showed a 24 h transcription profile").

Response: We have made the suggested change to the manuscript.

5) While most of the manuscript talks of "wild type parasites" (for the controls in the sr10 KO experiments), some text in the methods, figure legends and figures only use the term "wild". This should be correct (put "wild type" everywhere).

Response: Corrected

6) Line 494: As far as I know, all 3 references (46-48) are in mice. To fit with the sentence "These processes are considered central (...) in other taxa.", including additional references backing the statement but in other taxa than mice, would be good.

Response: Two additional references have been added (49-50 in the revised manuscript) that cover the same processes in ants and Arabidopsis.

7) Line 715: "absence" instead of "absent".

Response: Corrected

8) Lines 747-750: Please explain how HSP40 was checked to be non-rhythmic in *P. Chabaudi* (and thus, suitable to be used as a control gene for rhythmic gene expression analyses), and similarly, provide evidence that Seryl tRNA ligase is non-rhythmic in *P. falciparum*.

Response: We have chosen HSP40 to normalize the transcript expression because it was found to be non-rhythmic in expression (expressed constantly throughout the IDC) in all the time-series RNAseq data from all the strains used in this study including host rhythm matched and mismatched *P. chabaudi* parasites. Similarly, Seryl tRNA ligase was also found to be non-rhythmic in our IDC time-series RNAseq data. Seryl tRNA ligase has also been widely used as a control gene in previous *P. falciparum* studies. We have added this information to the revised manuscript.

Reviewer #2 (Remarks to the Author):

I am happy with the rebuttal letter and revised manuscript.

Reviewer #3 (Remarks to the Author):

The revised manuscript is significantly improved and all of my concerns have been adequately addressed.